# ASIDE: Adaptive and Separable Interventional Dynamics via Progressive Meta-Learning

## Abstract

To decide how to change the future trajectories of a dynamics system, it is important to predict not only the intrinsic dynamics of the system but also its response to external interventions. While notable progress has been made in learning intervention effects over time, existing research has prioritized the challenge of time-varying confounding in observational data. Significant challenges however remain in aspects related to the modeling and inference of latent dynamics. A first and foremost challenge lies in the need to separate, from a composite observation, the natural temporal evolution of intrinsic dynamics from its response to external interventions. This challenge is further exacerbated by the need to integrate rich history information into these latent dynamics. In this paper, we present a novel framework of adaptive and separable interventional dynamics (ASIDE) to overcome these challenges. First, we decompose the latent dynamics into separate components of intrinsic dynamics and its responses to external interventions at the latent space. This is in contrast to existing approaches that model and infer the composite dynamics as a black box. Second, we leverage meta-learning to enable these components to separately adapt to their relevant context examples in past history, addressing both inter- and intra-subject variabilities. This is in contrast to existing approaches that use history only to initialize a *one-size-fit-all* forecasting function. On synthetic and real benchmarks, we demonstrate the advantage of ASIDE in improving forecasting accuracy for both intrinsic and interventional dynamics, in settings with or without time-varying confounding.

## 1 Introduction

Across diverse domains, high-dimensional time-series observations are becoming increasingly abundant. This trend underscores the growing importance of time-series modeling as a foundation for enabling prediction and optimal control of observed systems (Krishnan et al., 2015). While forecasting the *intrinsic dynamics* native to a system is important for predicting its future trajectories, to inform optimal decisions that can influence such trajectories requires predicting the effect of *external interventions* on the system's intrinsic dynamics (Krishnan et al., 2015; Gwak et al., 2020). Using medicine as an example, longitudinal multi-modal data of an individual can be leveraged to predict the progression of an underlying health condition, while the decision of what interventions (*e.g.*, medication, life-style, etc.) to best improve such progression requires an ability to model and predict the effect of these interventions on the individual's native health condition over time.

Significant advances have been made in developing deep learning models for modeling the latent dynamics underlying high-dimensional time-series data (Chung et al., 2015; Krishnan et al., 2015; Fraccaro et al., 2017; Botev et al., 2021). However, most of these developments are focused on the intrinsic dynamics of a system, with limited consideration about the effects of external interventions (Krishnan et al., 2015; Gwak et al., 2020; Brouwer et al., 2022). In parallel, there has been a rising interest in modeling intervention effects over time from observational data: however, existing works have prioritized the challange of time-varying confounding, leveraging established techniques for dynamics mdoeling such as LSTM (Lim, 2018; Bica et al., 2020; Berrevoets et al., 2021), transformers (Melnychuk et al., 2022), and neural ordinary differential equations (ODEs)

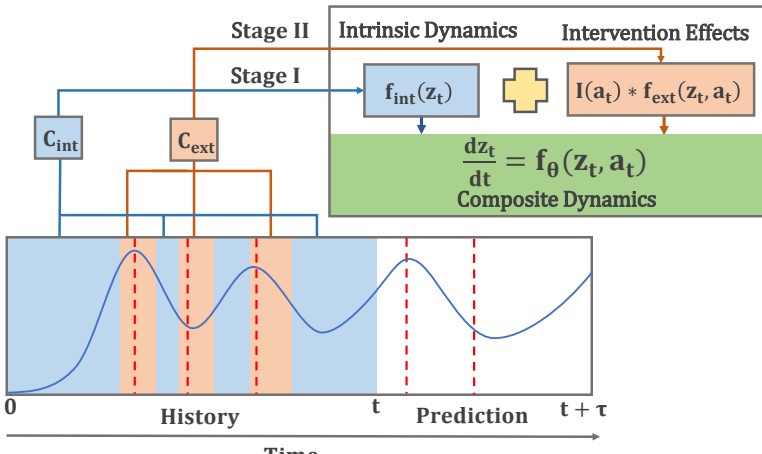

Figure 1: ASIDE with separate modeling of intrinsic dynamics and its response to external interventions, separately adaptive from context samples in the history via progressive meta-learning.

(Brouwer et al., 2022). At the intersection of these two vibrant research areas, significant gaps remain in the modeling and inferrence of latent dynamics under external interventions.

A first and foremost challenge lies in the need to separate, from a composite observation, the natural temporal evolution of intrinsic dynamics from its response to external interventions. Consider time-series covariate data $\{\mathbf{x}_t\}_{t=1}^T$ and its corresponding latent dynamics states $\{\mathbf{z}_t\}_{t=1}^T$. Assume that the temporal evolution of $\mathbf{z}_t$ under external interventions $\mathbf{a}_t$ is governed by a dynamics function $\frac{d\mathbf{z}_t}{dt} = f(\mathbf{z}_t, \mathbf{a}_t)$, which describes both how $\mathbf{z}_t$ naturally evolves (*intrinsic dynamics*) and its response to the external intervention $\mathbf{a}_t$. Unfortunately, these two mechanisms are not separately observed. Existing works in longitudinal intervention-effect modeling partially addresses this challenge from the perspective of causal-inference, *e.g.*, by extracting from the history a latent representation $\mathbf{z}_t$ that is minimally predictive of $\mathbf{a}_t$ when predicting the effect of $\mathbf{z}_t$ and $\mathbf{a}_t$ on an outcome variable $\mathbf{y}_{t+1}$. These however do not explicitly disentangle intrinsic dynamics and its response to external interventions in $f(\mathbf{z}_t, \mathbf{a}_t)$. Limited works have attempted this disentangling. In (Gwak et al., 2020), two separate neural ODEs were used to respectively model intrinsic dynamics $f(\mathbf{z}_x)$ and its effect from external interventions $f(\mathbf{z}_a)$. In (Brouwer et al., 2022), the latent dynamics is modeled as a neural controlled differential equation (CDE) where the dynamics $f(\mathbf{z}_t)$ is modulated by incoming treatments $\mathbf{a}_t$'s. Unfortunately, simply separating the components in the formulation of $f(\mathbf{z}_t, \mathbf{a}_t)$ does not guarantee that these components can be effectively separated from data during inference.

Secondly, consider $\mathbf{z}_{t+1:t+\tau}$ generated by $f(\mathbf{z}_t, \mathbf{a}_t)$ over any time window $\tau$ and its associated co-variates $\mathbf{x}_{t+1:t+\tau}$ and outcomes $\mathbf{y}_{t+1:t+\tau}$. Existing dynamic intervention-effect models attempt to describe this by a good initial estimate $\hat{\mathbf{z}}_t$ (from history) along with a *global* dynamics function that applies to all data samples (Lim, 2018; Bica et al., 2020; Melnychuk et al., 2022; Seedat et al., 2022; Wang et al., 2024a). This is typically achieved in a two-stage encoding-decoding framework (Lim, 2018; Bica et al., 2020; Seedat et al., 2022; Hess & Feuerriegel, 2025): in the first stage, a sequential encoder is trained to extract a latent representation $\hat{\mathbf{z}}_t$ from past history; in the second stage, a fore-casting decoder is then trained to use $\hat{\mathbf{z}}_t$ to forecast ahead for a window length of $\tau$. This popular approach has two limitations. First, the training of the encoder, *i.e.*, the extraction of $\hat{\mathbf{z}}_t$, is unaware of the primary forecasting objective in the second stage. Second and more importantly, a global $f(\mathbf{z}_t, \mathbf{a}_t)$ can be limited in its ability to describe the heterogeneity in both the intrinsic dynamics and responses to external interventions – a variability that can exist both among individual systems (*e.g.*, different patient subgroups) and within the same system over time (*e.g.*, different disease stages of the same patient). While $\hat{\mathbf{z}}_t$ as an estimated initial condition may capture such heterogeneity from past history, the ability for such information to pass forward in a *one-size-fit-all* dynamics function is not clear, especially as the forecasting horizon increases.

In this paper, we overcome these two challenges in a novel framework of ASIDE to achieve adaptive and separable interventional dynamics. As outlined in Fig. 1, ASIDE has three major innovations.

- We separately model the intrinsic dynamics $f_{int}(\mathbf{z}_t)$ and its response to external interventions $f_{ext}(\mathbf{z}_t, a_t)$ within $f(\mathbf{z}_t, a_t)$. This shares the motivation for (Gwak et al., 2020) and is in contrast to existing approaches that learn $f(\mathbf{z}_t, a_t)$ as a composite black-box.

- We further allow both dynamics to vary with *context* embeddings extracted from past history, enabled in a novel end-to-end meta-learning framework to allow rapid feed-forward extraction of context embeddings and adaptive forecasting that address both inter- and intra-subject variations. This is in contrast to existing approaches that uses history information to extract only the initial condition for a global *one-size-fit-all* dynamics function.

- With a novel integration of the meta-inference and separate-modeling strategies, we eventually allow $f_{int}(\mathbf{z}_t; \mathbf{c}_{int})$ and $f_{ext}(\mathbf{z}_t, a_t; \mathbf{c}_{ext})$ to separately adapt to the context embedding $\mathbf{c}_{int}$ and $\mathbf{c}_{ext}$ extracted from their relevant parts of the history data: *i.e.*, intrinsic dynamics is adapted from intervention-free segments of the history, and interventional dynamics are adapted from the contrast of interventional to intervention-free segments (synthesize or factual) ) of the history. This separation is further enhanced by a progressive learning strategy to first estimate $f_{int}$ and, with which, to isolate $f_{ext}$ from their composite observations. We show that this integration–*adaptive and separable dynamics—is the key for* improving forecasting accuracy, especially in long forecasting horizons and increasing heterogeneity.

We first evaluated ASIDE in a synthetic benchmark of tumor growth under radiation and chemotherapies (Geng et al., 2017): to isolate the effect of the dynamics modeling strategies introduced in ASIDE, we considered data settings with and without treatment assignment bias. We then evaluated ASIDE on a real dataset from MIMIC-III (Johnson et al., 2023), which are inherently associated with unknown treatment assignment bias inherent in observational data. In both settings, we demonstrated the improved performance of ASIDE for time-series forecasting under dynamic external interventions in comparison to contemporary baselines.

## 2 RELATED WORKS

**Modeling intervention effects over time:** Rapid progress has been made in learning intervention effects over time (Lim, 2018; Bica et al., 2020; Melnychuk et al., 2022; Seedat et al., 2022; Berrevoets et al., 2021; Seedat et al., 2022; Hess & Feuerriegel, 2025; Vanderschueren et al., 2023; Wang et al., 2024a). Most works prioritize the challenge of time-varying confounding and treatment assignment bias, represented by propensity weight (Lim, 2018) and invariant representation (Bica et al., 2020; Melnychuk et al., 2022; Seedat et al., 2022) approaches. In terms of dynamics modeling, earlier works have mostly adopted a two-stage learning process, where the first-stage trains an *encoder* to extract latent representations $\mathbf{z}_t$ from past history $\mathcal{H}_{1:t}$ and, with this fixed, a second-stage *decoder* learns to take the encoded $\mathbf{z}_t$ to predict treatment outcome given intervention $\mathbf{a}_t$ within a horizon of $\tau$. Different types of neural architectures have been used, such as recurrent neural networks (Lim, 2018; Bica et al., 2020) and CDEs (Seedat et al., 2022; Hess & Feuerriegel, 2025; Vanderschueren et al., 2023). Recognizing the limitations associated with such two-stage training, especially that the encoding from history is not made aware of the forecasting objectives, growing recent works have attempted learning the encoding-decoding process end-to-end, using neural ODEs (Brouwer et al., 2022) and transformers (Melnychuk et al., 2022; Wang et al., 2024a).

Orthogonal to the contribution of these existing works, ASIDE aims to advance intervention-effect modeling by enabling the learning of separable and adaptive interventional dynamics.

**Separating intrinsic dynamics and interventional effects:** There have been limited works that share our motivation in separating the intrinsic and interventional dynamics from observed time-series. In (Gwak et al., 2020), this is achieved by using two separate neural ODEs which are then combined in a third neural ODE to generate the observed composite trajectories. In (Seedat et al., 2022), the latent dynamics is modeled as a neural CDE where the intrinsic dynamics $f(\mathbf{z}_t)$ is modulated by incoming treatments $\mathbf{a}_t$'s. This explicit separation of intrinsic dynamics and intervention effects improves the interpretability of the latent dynamics models compared to a black box. However, effective inference strategies to ensure such separation remain an open problem. Furthermore, neither of these works have considered adaptive latent dynamics.

**General time-series modeling:** Beyond the literature of intervention effect modeling over time, time-series modeling span a vast space ranging from uni/multi-variate time-series forecasting (Liu et al., 2024; Wang et al., 2024b), latent dynamics modeling for time series of high-dimensional data (Chung et al., 2015; Krishnan et al., 2015; Fraccaro et al., 2017; Botev et al., 2021), and even causal time-series models (Yao et al., 2022; 2021; Song et al., 2024). An important distinction for dynamic intervention effect modeling arises from 1) the need to consider external control (*i.e.*, intervention) over time, and 2) the need to accommodate an increasing/variable-length history (of covariates, treatment, and outcome). In contrast, the other aforementioned areas of time-series modeling typically do not consider external control inputs. More importantly, they either consider a fixed-length look-back window for prediction (Liu et al., 2024; Wang et al., 2024b; Chung et al., 2015; Krishnan et al., 2015; Fraccaro et al., 2017; Botev et al., 2021) or, in the case of most causal time-series models, considers reconstruction of a fixed-length time series. They are thus not directly extendable to intervention effect modeling over time without customized design.

## 3 METHODOLOGY

As outlined in Fig. 1, ASIDE includes three key innovations: separable models of the intrinsic and interventional dynamics at the latent space, meta-learning to adapt latent dynamics to context examples in past history, and their integration to eventually realize the separate adaptation of intrinsic and interventional dynamics from their relevant context in the history. Below, we describe ASIDE by its adaptable latent dynamics models (Section 3.1), the extraction of context embedding from history to adapt the dynamics (Section 3.2), and the progressive meta-learning scheme (3.3).

### 3.1 GENERATION PROCESS: ADAPTIVE AND SEPARABLE LATENT DYNAMICS

While our model is agnostic to the type of functions used to describe latent dynamics, in this paper we choose a neural ODE to describe the latent dynamics as a *continuous* process:

$$\frac{d\mathbf{z}_t}{dt} = f_\theta(\mathbf{z}_t, \mathbf{a}_t) = f_{\theta_{\text{int}}}(\mathbf{z}_t; \mathbf{c}_{\text{int},t}) + \sum_k \mathcal{I}^k(a_t^k) f_{\theta_{\text{ext}}^k}(\mathbf{z}_t, a_t^k; \mathbf{c}_{\text{ext},t}^k); \ \mathbf{y}_{t+1} = g_\eta(\mathbf{z}_t, \mathbf{a}_t) \tag{1}$$

where $f_{\theta_{\text{int}}}$ models the intrinsic dynamics parameterized by $\theta_{\text{int}}$. Multiple interventions can exist, where $a_t^k \in \mathbf{a}_t$ represents intervention $k$ and the indicator function $\mathcal{I}(a_t^k) = 1$ flags its presence at time $t$. The response of $\mathbf{z}_t$ to intervention $k$ is modeled by $f_{\theta_{\text{ext}}^k}$ parameterized by $\theta_{\text{ext}}^k$. $g$ describes the effect of $\mathbf{z}_t$ and $\mathbf{a}_t$ parameterized by $\eta$. Instead of learning fixed functions for $f_{\theta_{\text{int}}}$ and $f_{\theta_{\text{ext}}^k}$'s, we allow them to change with time-varying embeddings $\mathbf{c}_{\text{int},t}$ and $\mathbf{c}_{\text{ext},t}^k$'s that are separately identified from history $\mathcal{H}_{1:t} = \{\mathbf{x}_{1:t}, \mathbf{a}_{1:t-1}, \mathbf{y}_{1:t}\}$ up to time t. While different adaptation mechanisms exist (Jayakumar et al., 2020), here we consider a simple conditioning for additive adaptation.

The generation process as described in equation 1 differ from existing works in two aspects. First, the latent dynamics is explicitly decomposed into intrinsic dynamics and its response to interventions. Second, latent dynamics functions are adaptable instead of fixed for all training samples.

### 3.2 INFERENCE PROCESS: ADAPTATION VIA DYNAMICS-SPECIFIC CONTEXT EMBEDDING

To optimize the generation process in equation 1, existing works (Lim, 2018; Bica et al., 2020; Melnychuk et al., 2022; Seedat et al., 2022; Wang et al., 2024a) mostly focus on first learning an *encoder* to extract a latent representation $\mathbf{z}_t$ from the history $\mathcal{H}_{1:t}$, which is then utilized to initialize a fixed forecasting function optimized by predicting forward for a duration of $\tau$. ASIDE differs in the following aspects. First, because the trajectory of $\mathbf{z}_t$'s is *governed* by $f_\theta$, we shift the focus of learning to $f_\theta$: this results in a fundamentally-different inference formulation where information from a patient's history $\mathcal{H}_{1:t}$ is used to adapt $f_\theta$ to capture inter- and intra-subject variability, optimized by forecasting forward for a window of $\tau$ using a simple $\mathbf{z}_t$ estimated from recent observations. Second, to encourage the separation of intrinsic dynamics and responses to interventions, we design the extraction of $\mathbf{c}_{\text{int},t}$ and $\mathbf{c}_{\text{ext},t}^k$ from 1) different portions of the history data $\mathcal{H}_{1:t}$ depending on their relevance to $f_{\text{int}}$ and $f_{\text{ext}}$ and 2) via different extraction mechanisms.

**Inference of initial latent states:** To shift the learning focus to a strong forecasting function $f_\theta$ that is able to capture history information, we use a simple strategy to infer $\mathbf{z}_t$ from the first frames of $\mathbf{x}_{t-l-1:t-1}(l < \tau)$ via a neural encoder $\Psi_{\phi_z}$ with weight parameters $\phi_z$: $\hat{\mathbf{z}}_t = \Psi_{\phi_z}(\mathbf{x}_{t-l-1:t-1})$.

**Inference of intrinsic dynamics:** To infer $\mathbf{c}_{\text{int},t}$ for adapting $f_{\theta_{\text{int}}}$ at time $t$, we assume that it may be shared by past trajectories of natural dynamics of the same individual. To remove confounding interventional effects in past history, we further consider only the trajectories in history $\mathcal{H}_{1:t}$ that are free from any intervention, denoted by $\mathcal{H}_{1:t} * \mathcal{I}_{1:t}^0$ with $\mathcal{I}_{1:t}^0$ indicating the absence of intervention ($= 1$) or not ($= 0$) at each time instant $t$. We further divide available $\mathcal{H}_{1:t} * \mathcal{I}_{1:t}^0$ into $l_{\text{int}}$ number of segments $\mathbf{s}_{\text{int}}$'s of duration $\tau$, where $l_{\text{int}}$ varies as the history grows. Each individual segment $\mathbf{s}_{\text{int}}$ is fed into an encoder $\Psi_{\phi_{\text{int}}}(\cdot)$ to extract an embedding, which are then aggregated across segments as:

$$\mathbf{c}_{\text{int,t}} = \mathcal{M}_{\text{int}}(\mathcal{H}_{1:t} * \mathcal{I}_{1:t}^0) = \frac{1}{l_{\text{int}}} \sum\nolimits_{\mathbf{s}_{\text{int}} \in \mathcal{H}_{1:t} * \mathcal{I}_{1:t}^0} \Psi_{\phi_{\text{int}}}(\mathbf{s}_{\text{int}}) \tag{2}$$

where we adopt a simple averaging here to extract the shared embedding by the context samples $\mathbf{s}_{\text{int}} \in \mathcal{H}_{1:t} * \mathcal{I}_{1:t}^0$. Additional weighted averaging or time decay can be added to share with only recent history, or using attention mechanism to find similar dynamics in the history.

**Inference of response to external interventions:** The inference of $\mathbf{c}_{\text{ext},t}^k$ for adapting $f_{\theta_{\text{ext}}}$ is more challenging as, unlike intrinsic dynamics, the history will never have *intrinsic-free* trajectories. Instead, for any intervention $k$, only its composite effect with intrinsic dynamics can be observed. To separate out the latter, we leverage the concept of *counterfactuals*: for any factual composite trajectory under the effect of intervention $k$, we introduce an *intervention-free* counterfactual for an encoder $\Psi_{\phi_{\text{ext}}}^k(\cdot)$ to compare and extract an embedding that accounts for the difference due to intervention $k$. This is realized in two strategies. First, for each segment $\mathbf{s}_{\text{ext}}^k \in \mathcal{H}_{1:t} * \mathcal{I}_{1:t}^k$ with $\mathcal{I}_{1:t}^k$ indicating the presence of intervention type $k$ ($= 1$) or not ($= 0$), we synthesize its counterfactual $\mathbf{s}_{\text{ext,CF}}^k$ with our learned $f_{\theta_{\text{int}}}$ and the initial latent state estimate, projecting what the segment would have looked like if intervention $k$ was not applied. The pair of factual and counterfactual samples are fed into an encoder $\Psi_{\phi_{\text{ext}}^k}(\cdot)$ to extract an embedding, which are then aggregated across the segments:

$$\mathbf{c}_{\text{ext,t}}^k = \mathcal{M}_{\text{ext}}^k(\mathcal{H}_{1:t} * \mathcal{I}_{1:t}^k) = \frac{1}{l_{\text{ext}}^k} \sum\nolimits_{\mathbf{s}_{\text{ext}}^k \in \mathcal{H}_{1:t} * \mathcal{I}_{1:t}^k} \Psi_{\phi_{\text{ext}}^k}(\mathbf{s}_{\text{ext}}^k, \mathbf{s}_{\text{ext,CF}}^k) \tag{3}$$

where $l_{\text{ext}}^k$ represents the number of history segments with intervention $k$. Similarly, a simple averaging is used here, although more advanced aggregation strategy can be used depending on prior knowledge about the intra-subject variability in an individual's response to intervention $k$.

Alternatively, we can include factual intervention-free segments in the history $\mathcal{H}_{1:t} * \mathcal{I}_{1:t}^0$ in addition to the interventional segments $\mathcal{H}_{1:t} * \mathcal{I}_{1:t}^k$, along with the mask $\mathbb{I}(a_k = 1)$ that indicates the presence or absence of $a_k$. While not paired, this can be considered as comparing interventional and intervention-free data at a distribution level. We realize this with an convolutional architecture over these segments concatenated with intervention masks:

$$\mathbf{c}_{\text{ext,t}}^k = \mathcal{M}_{\text{ext}}^k([\mathcal{H}_{1:t} * \mathcal{I}_{1:t}^k, \mathcal{H}_{1:t} * \mathcal{I}_{1:t}^0; \mathbb{I}(a_k = 1)]) \tag{4}$$

### 3.3 Progressive Meta-Leaerning

**Learn-to-adapt meta-objectives:** Given a dataset consisting of $N$ unique time-series, we consider the forecasting task of predicting the values of $\mathbf{y}_{t+1:t+\tau}^i$ given the values of history to the point $\mathcal{H}_{1:t}^i = \{\mathbf{x}_{1:t}^i, \mathbf{a}_{1:t-1}^i, \mathbf{y}_{1:t}^i\}$ and intervention assignment $\mathbf{a}_t^i$, where $i = 1 : N$ indicates the $i$-th time-series in the dataset. For each $\mathbf{y}_{t+1:t+\tau}^i$, the predicted $\hat{\mathbf{y}}_{t+1:t+\tau}^i$ is generated as described by equation 1, with $\mathbf{z}_t^i$ estimated by the initial state encoder, and $f_{\theta_{int}}$ and $f_{\theta_{ext}^k}^k$'s respectively adapted by $\mathcal{M}_{\text{int}}$ and $\mathcal{M}_{\text{ext}}^k$'s as described in equation 2 – equation 4. The mean-squared-error (MSE) loss between $\hat{\mathbf{y}}_{t+1:t+\tau}^i$ and $\mathbf{y}_{t+1:t+\tau}^i$ is used to optimize the weight parameters of the latent dynamics functions $\theta_{\text{int}}$ and $\theta_{\text{ext}}^k$'s, their corresponding encoders for adaptation $\phi_{\text{int}}$ and $\phi_{\text{ext}}^k$'s, along with that for the initial state encoder $\phi_z$ and emission function $\eta$.

$$\min_{\phi_{\text{ext}}^k, \phi_{\text{int}}, \phi_z, \theta_{\text{int}}, \theta_{\text{ext}}^k, \eta} \sum_{i=1}^{N} \sum_{t=1}^{T-\tau} \|\mathbf{y}_{t+1:t+\tau}^i - \hat{\mathbf{y}}_{t+1:t+\tau}^i(\mathcal{H}_{1:t}, \mathbf{a}_t)\|_2^2 \quad k = 1, \cdots, K \tag{5}$$

where $K$ is the maximum numbers of intervention types considered in the dataset.

The optimization of equation 5 thus corresponds to a meta-learning objective when treating the prediction of each $\mathbf{y}_{t+1:t+\tau}^i$ as its own task, with context samples selected from the history $\mathcal{H}_{1:t}$ as described in Section 3.2, to adapt the intrinsic dynamics $f_{\theta_{\text{int}}}$ and interventional dynamics $f_{\theta_{\text{ext}}}^k$'s to the specific task. $\mathcal{M}_{\text{int}}$ and $\mathcal{M}_{\text{ext}}^k$ as described in equation 2 – equation 4 thus represent feedforward meta-models to extract task-specific embedding for rapid adaptation of latent dynamics models.

**Progressive meta-learning to separate intrinsic and interventional dynamics:** To further ensure the separation of the intrinsic and multiple interventional dynamics we adopt a progressive training strategy where different componets of the model are estimated at a schedule. More specifically, we first optimize $\phi_{\text{int}}$ and $\theta_{\text{int}}$ related to intrisinc dynamics, *i.e.*, the intrinsic dynamics function $f_{\theta_{\text{int}}}$ and the meta-encoder $\mathcal{M}_{\text{int}}$ used to adapt it. Note that this optimization only involve intervention-free observations, removing the challenge of separating composite effects. With the optimized $\phi_{\text{int}}$ and $\theta_{\text{int}}$ fixed, we then simultaneously optimize $\phi_{\text{ext}}^k$'s and $\theta_{\text{int}}^k$'s related to responses to interventions, *i.e.*, interventional dynamics functions $f_{\theta_{\text{ext}}^k}$'s and the meta-encoders $\mathcal{M}_{\text{int}}^k$ used to adapt them, for all intervention types $k$. While this stage of the training does involve interventional data with composite effects, leveraging the optimized intrinsic dynamics facilitates the separation of interventional dynamics. Finally, all model parameters are finetuned together as a fully-integrated model. The initial state encoder $\Psi_{\phi_z}$ and emission $g_\eta$ are trained throughout. This progressive training is achieved by turning off the gradient flow to the parameters not involved in training at different stages, each with its respective optimization and learning hyperparameters. Transition between the stages is determined by when the loss plateaus in a stasge or when a max epoch limit is reached.

## 3.4 RELATION TO IDENTIFIABILITY THEORY

Consider the outcome $\mathbf{y}_{t+1:t+\tau}$ of interest to be generated from latent intrinsic and interventional dynamics governed by some *true* parameters $\mathbf{c}$'s, *e.g.,* the tumor growth and radiation/chemotherapy effect parameters in governing the trajectory of tumor volume. From the point of view of a generative model, it is already estsablished that the equivalance at the observation of $p(\mathbf{y}_{t+1:t+\tau})$ does not translate to the identifiability of the latent $\mathbf{c}$ without auxiliary information (Hyvarinen et al., 2019; Khemakhem et al., 2020). Interestingly, in a recent work (Ye et al., 2024), meta-learning is established as a novel solution to construct a conditional generative model $p(\mathbf{y}_{t+1:t+\tau}|\mathbf{u}) = \int p(\mathbf{y}_{t+1:t+\tau}|\mathbf{c})p(\mathbf{c}|\mathbf{u})d\mathbf{c}$, such that the identifiability of $\mathbf{c}$ can be achieved by leveraging auxiliary information $\mathbf{u}$ in the form of context samples. The solution of ASIDE represents a deterinistic counterpart to this probablistic generative model, where the context embedding $\mathbf{c}_{\text{int}}$ and $\mathbf{c}_{\text{ext}}$'s correspond to $\mathbf{c}$ conditioned on context samples in the history. This indicates that the meta-learning formulation underpinning ASIDE, in addition to enabling adaptive and separate dynamics, may also facilitate the identification of the true parameters of the latent intrinsic and interventional dynamics, which in turn determine part of the causal mechanisms underlying the outcome of interest. Due to the deterministic form of ASIDE, we leave a rigorous theoretical probe to future works, but will examine identifiability metrics for $\mathbf{c}_{\text{int}}$ and $\mathbf{c}_{\text{ext}}$'s for empirical evidence of this theoretical insight.

## 4 EXPERIMENTS

Counterfactual outcomes are not commonly observed for real-world data, due to which synthetic data have become important for evaluating intervention effect models (Lim, 2018; Bica et al., 2020; Seedat et al., 2022; Melnychuk et al., 2022). We thus first evaluated ASIDE on the well-established benchmark generated by a pharmacokinetic-pharmacodynamic (PK-PD) model of tumor growth in lung cancer patients that includes the effects of chemotherapy and radiotherapy (Geng et al., 2017). To isolate the effect of dynamics models, we considered experimental settings with and without time-varying confoudning due to treatment assignment bias. To test feasibility in real world settings, we then conducted experiments on the MIMIC-III (Johnson et al., 2023), an electronic health record dataset with inherent real-world treatment assignment bias in observational data (Bica et al., 2020).

**Baselines:** On both datasets, we considered representative baselines in intervention effect modeling over time, including: 1) MSM (Hernán et al., 2000) 2) RMSN (Lim, 2018), 3) CRN (Bica et al., 2020), 4) CausalTransformers (CT) (Melnychuk et al., 2022), 5) SCIP (Hess & Feuerriegel, 2025), 6) TE-CDE (Seedat et al., 2022) 7) TESAR-CDE (Vanderschueren et al., 2023) 8) ACTIN (Wang et al., 2024a). Among these, MSM represents the classical causal models not based on neural networks. For the rest, RMSN and CRN rely on two-stage encoder-decoder training using RNNs. TE-CDE, SCIP and TESAR-CDE use two-stage Controlled Differnetial Equations (CDEs). CT and ACTIN are based on end-to-end transformer architectures.

**Metrics:** We evaluated the performance of all models by their accuracy in predicting counterfactual outcomes over time, measured by rooted-mean-square-error (RMSE). Following standard

| Heterogeneity | Intrinsic Growth ($\rho$) | | Radio effect ($\alpha$) | | Chemo effect ($\beta_c$) | |
|---|---|---|---|---|---|---|
| | lower | upper | lower | upper | lower | upper |
| 0 | $7 \times 10^{-5}$ | $7 \times 10^{-5}$ | .0398 | .0398 | .028 | .028 |
| 1 | 0.0 | $7 \times 10^{-3}$ | 0.0 | .208 | .0273 | .0287 |
| 2 | 0.0 | $7 \times 10^{-3}$ | 0.0 | .508 | .0259 | .0301 |
| 3 | 0.0 | $21 \times 10^{-3}$ | 0.0 | .508 | .0259 | .0301 |

Table 1: Parameter ranges for different levels of heterogeneity in data generation.

practice, the RMSE is normalized by the maximum volume of the tumor (death threshold defined in Lim (2018)) fon the synthetic dataset. To glimpse into the potential benefit ASIDE in identifying the true parameters in the data generation equation, in a subset of experiments, we further evaluated the mean correlation coefficient (MCC) metric following (Khemakhem et al., 2020) to measure the identified $\mathbf{c}_{int}$ and $\mathbf{c}_{ext}$'s against the true data-generating intrinsic & interventional dynamics parameters.

### 4.1 SYNTHETIC DATA EXPERIMENTS

Following the PK-PD model in (Geng et al., 2017) for non-small cell lung cancer, we used the following Gompertz model to describe the grwoth of tumor volume with a starting volume of $V_0$:

$$V_{t+1} = V_t(1 + \rho \log(\tfrac{K}{V}) - \beta_c C_t - (\alpha r_t + \beta r_t^2) + \epsilon) \tag{6}$$

where parameters $\rho$ and $K$ control natural growth dynamics, $\beta_c$ controls the effect of chemotherapy with dose $C_t$, and $\beta, \alpha$ control the effect of radiotherapy dose of $r_t$. The parameter $K$ is the carrying capacity of the model. More details are provided in Appendix B.1

**Treatment assignment bias:** To isolate the benefits of the dynamics modeling strategies introduced by ASIDE, we first considered random treatment assignment with a probability of $p_0 = 0.1$ regardless of tumor volume: the relatively low assignment probability was chosen to avoid generating numerous time series where the tumor was killed within a small time window. To include treatment assignment bias, we then adopted the mechanism for treatment assignment in (Melnychuk et al., 2022): $p(a_t) = p_0 \, \sigma(\gamma(\tfrac{V_t}{V_{max}} - 1/2)$ with $p_0 = 0.2$, considering increasing levels of treatment bias as controlled by $\gamma = 1, 2, 3, 4$. For a fair comparison, for all baselines that use treatment-invariant representations to address confounding, we disabled their adversarial training element when there wsa no confounding. For ASIDE, we tested it with and without the additional component of treatment-invariant representations when considering time-varying confounding.

**Heterogeneity levels:** To examine the importance of *adaptive* latent dynamics, we created training data with four different levels of heterogeneity in the key parameters ($\rho, \beta_c, \alpha, \beta$) in Equation (6) by controlling their sampling to be within a preset range from the mean as summarized in Table 1.

**Prediction horizon:** Compared to existing works that integrate history into an initial condition for forecasting in time, ASIDE's use of history to adapt the dynamics function is expected to carry heterogeneity information forward for a longer prediction horizon. To delineate this benefit, we considered prediction horizons of lengths longer than most existing studies ($\tau = 5, 10, 15, 20$). Within a given prediction horizon, we also examined the per-step prediction RMSE over time.

#### 4.1.1 RESULTS WITHOUT CONFOUNDING

Figure 2 summarizes the results for the normalized RMSE for all models considered, under different levels of heterogeneity and over different prediction horizons when there was no treatment assignment bias: note that TE-CDE and SCIP were removed from Figure 2 because their RMSEs were at a much larger magnitude compared to the other models, but their complete results can be found in Table 8 in Section C.1 of the Appendix. As shown, ASIDE was able to provide significant margins of improvement over the included baselines across all prediction horizons and all levels of heterogeneity. As further highlighted in Fig. 3(a), this improvement overall improved as $\tau$ increased for all levels of heterogeneity. Fig. 3(b) further shows the per-step RMSE prediction accuracy of

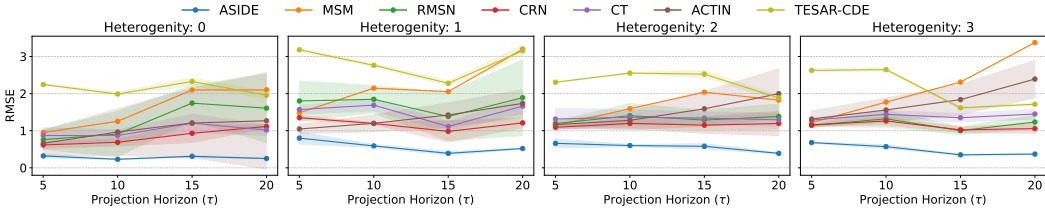

Figure 2: Test RMSE for the baselines and ASIDE over increasing projection horizons, at different levels of data heterogeneity. TECDE and SCIP are not included in the plots due to a much higher value of RMSE. Their full quantitative results are included in Appendix C.1 along with all baselines.

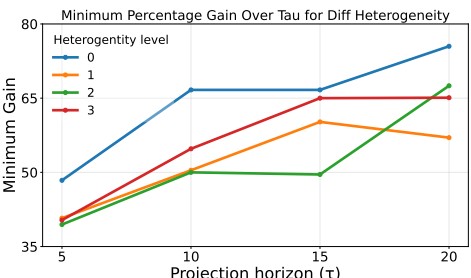

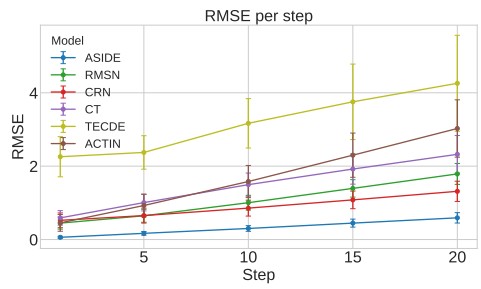

(a) Percentage gain over the second-best model across $\tau$ for different heterogeneity levels

(b) Stepwise RMSE for all the models when trained for heterogeneity level 3 and $\tau = 20$

Figure 3: (a) ASIDE's gain increases over baselines as $\tau$ increases. (b) ASIDE not only has the lowest RMSE but also the slowest trend of deterioration over the forecasting horizon.

ASIDE when trained to predict for a horizon of $\tau = 20$ at the highest level of heterogeneity, where ASIDE not only consistently attained the lowest prediction error but also the slowest deterioration over time *versus* all baseline models. This provided strong evidence that 1) ingesting history into the latent dynamics is stronger than ingesting history into an initial condition for predicting over longer horizons, and 2) adaptive dynamics is important for addressing data heterogeneity.

To further demonstrate the benefits of separable dynamics enabled by ASIDE, we examined the RMSE for intervention-free and interventional segments of the test samples separately. As shown in the example in Table 9, ASIDE delivered significantly improved RMSE for both predicting intrinsic dynamics and its changes under radiation or chemotherapies. While most baseline methods experienced an increase in error when predicting the effect of radiation therapies, potentially due to its relatively small effect compared to the growth of tumor due to intrinsic dynamics, ASIDE was able to maintain consistently prediction errors across the various dynamics. Table 14 in Appendix C.4 summarizes the MCC metrics between the inferred context embeddings and the true parameters $\rho, \beta_c$, and $\alpha$ of the data generating equation (6) at heterogeneity level 1-3: heterogeneity level 0 was exlucded from this analysis because the identifiabilty requires sufficient variability of the parameters to be observed (Ye et al., 2024). These empirical results provided initial evidence that parameter $\rho$ for the intrinsic dynamics and $\alpha$ for radiation effect are well identified (MCC averaged at 0.79 across heterogeneity level 1-3). The poor identfiability of $\beta_c$ for the chemotherapy effect may point to invalidty of assumptions related to the injectivity of the mixing function (Ye et al., 2024) that can be investigatd in future works. Fig. 5 in Appendix C.2 provides examples of individual trajectories.

### 4.1.2 RESULTS UNDER TIME-VARYING CONFOUDING

Table 2 summarizes the RMSE of ASIDE *vs.* all baselines at varying levels of treatmetn assignment bias, for prediction horizon $\tau = 5$ and the highest heterogeneity level. Because the three CDE-baed variants differ primarily in the way they handle irregular time series and delivered similarly in-competitive performance in Section 4.1.1, in this experiment we only include their best representative TESCAR-CDE. As shown, even without any additional strategies for addressing time-varying confounding (denoted by $\alpha = 0$), ASIDE showed significantly higher accuracy than all baselines

| Models | $\gamma = 1$ | $\gamma = 2$ | $\gamma = 3$ | $\gamma = 4$ |
|---|---|---|---|---|
| CT | 1.66 (0.03) | 2.46 (0.13) | 3.72 (0.23) | 3.74 (0.24) |
| RMSN | 1.72 (0.20) | 2.24 (0.24) | 3.00 (0.35) | 3.20 (0.12) |
| CRN | 1.45 (0.04) | 2.10 (0.04) | 2.59 (0.03) | 2.76 (0.03) |
| MSM | 2.20 | 2.65 | 3.08 | 3.74 |
| TESAR-CDE ($\alpha$=0) | 2.98 (0.01) | 3.81 (0.06) | 4.88 (0.07) | 5.72 (0.07) |
| TESAR-CDE ($\alpha$=1) | 3.00 (0.03) | 3.88 (0.10) | 5.10 (0.14) | 5.90 (0.11) |
| ACTIN | 1.52 (0.17) | 1.73 (0.22) | 2.29 (0.28) | 2.67 (0.34) |
| AISDE ($\alpha$=0.0) | **0.93 (0.08)*** | 1.43 (0.09*) | **1.92 (0.22)*** | **2.33 (0.25)** |
| AISDE ($\alpha$=0.25) | 1.04 (0.24) | **1.37 (0.08)*** | 2.60 (0.67) | 2.40 (0.18) |
| AISDE ($\alpha$=0.50) | 0.96 (0.05) | 1.39 (0.05) | 2.29 (0.52) | 2.65 (0.68) |
| AISDE ($\alpha$=0.75) | 1.04 (0.31) | 1.71 (0.38) | 1.97 (0.14) | 2.52 (0.57) |
| AISDE ($\alpha$=1.0) | 1.01 (0.06) | 1.51 (0.13) | 1.96 (0.13) | 2.36 (0.15) |

Table 2: Test RMSEs obtained under different levels of treatment assignment bias on the synthetic tumor dataset (for $\tau = 5$ and heterogeneity level =3 ). Value of $\alpha$ indicates the strengths of the regularizer for treatment-invariant representations following (Bica et al., 2020). $*$ marks statistically significant difference over the best baseline based on paired-t tests ($p < 0.05$). For $\gamma = 4$, ASIDE was better than the second best model (ACTIN) at $p = 0.06$. MSM given the nature of the method did not rely on random seeds and thus no standard deivation is reported.

| Model | $\tau = 5$ | $\tau = 10$ | $\tau = 15$ | $\tau = 20$ |
|---|---|---|---|---|
| RMSN | 12.28 (1.88) | 12.58 (0.87) | 14.93 (1.40) | 14.09 (0.93) |
| CRN | 9.82 (0.10) | 10.46 (0.15) | 11.00 (0.12) | 11.46 (0.11) |
| CT | 9.66 (0.11) | 10.23 (0.15) | 10.58 (0.19) | 10.92 (0.22) |
| ACTIN | 9.94 (0.11) | 10.34 (0.14) | 10.65 (0.16) | 10.91 (0.19) |
| ASIDE | **9.55 (0.11)** | **10.02 (0.13)*** | **10.31 (0.15)*** | **10.56 (0.16)*** |

Table 3: Test RMSEs for different prediction projection horizon in MIMIC-III dataset. $*$ marks statistically significant difference over the best baseline based on paired-t tests ($p < 0.05$). For $\tau = 5$, the difference between ASIDE and the second-best model (CT) was at $p = 0.104$.

across all levels of confouding. The additional use of time-invariant representation, implemented through gradient reversal from a treatment prediction classifier following (Bica et al., 2020), introduced inconsistent gains to ASIDE at different confounding levels and with an overall statistically insignificant change – this observation was consistent to results reported in other works on this dataset (Melnychuk et al., 2022; Bica et al., 2020).

## 4.2 Real Data Experiments

MIMIC-III (Johnson et al., 2023) contains electronic health records of ICU patients. Following Melnychuk et al. (2022), we considered covariates $\mathbf{x}_t$ as 25 vitals and 3 static features, outcome $y_t$ as diastolic blood pressure, and treatment $\mathbf{a}_t$ as vasopressor and ventilation. We selected only the most competitive baselines in this experiment. Because of the treatment bias inheret within observational data, all baselines included their original mechanisms for addressing confounding.

Table 3 summarizes the test RMSE results. As shown, even without any specific mechanisms to addressing time-varying confounding, ASIDE was able to deliver an margin of improvement that was statistically significant over the second best models across all prediction horizons, except at $\tau = 5$ where $p = 0.104$ (*paired-t* tests). Similarly, while all models RMSE increased as $\tau$ increased, ASIDE demonstrated the least deterioration ($10\%$) compared to the baselines (ranging $13\%$ to $22\%$). Fig. 6 provides the per step RMSE over the prediction horizon.

## 4.3 Additional Ablation Results

To further understand the significant gain of ASIDE over considered baselines in its prediction accuracy, we conducted an ablation study on the key components of ASIDE: separated latent dynamics components, meta-learning for adaptive dynamics, and their integration as well as progressive learn-

| Model | Separate Dynamics | Meta-Learning | Progressive Training | RMSE (avg) |
|---|---|---|---|---|
| RMSN | ✗ | ✗ | | 1.14 (0.03) |
| CRN | ✗ | ✗ | | 1.16 (0.02) |
| CT | ✗ | ✗ | | 1.29 (0.06) |
| Model 1 (global joint) | ✗ | ✗ | ✗ | 1.09 (0.10) |
| Model 2 (global separate) | ✓ | ✗ | ✗ | 1.01 (0.04) |
| Model 3 (meta joint) | ✗ | ✓ | ✗ | 0.96 (0.02) |
| Model 4 (global separate prog) | ✓ | ✗ | ✓ | 0.97 (0.03) |
| Model 5 (meta separate) | ✓ | ✓ | ✗ | 0.72 (0.02) |
| ASIDE (meta separate prog) | ✓ | ✓ | ✓ | 0.68 (0.02) |

Table 4: Ablation results for different component models on $heterogenity = 3$ and $\tau = 5$

ing to ensure separation. This set of experiments was conducted considering only radiation therapy without the presence of chemotherapy: in the absence of the latter, the average tumor value in the dataset had increased which resulted in an increased RMSE in all models, as shown in Table 4 .

For ablation, we started with a global neural ODE (Model 1) in which a single global ODE function was learned at the latent space that was not conditioned on any history $\mathcal{H}$. This model delivered comparable performance to the other baselines considered. As we decomposed this single neural ODE into a formulation with separated intrinsic and interventional dynamics similar to Equation (1) but without meta-adaption (Model 2), a moderate improvement was obtained without statistical significance. Similarly, allowing this single model to adapt without separate dynamics (Model 3) also brought only a limited improvement, at the level similar to adding progressive training strategy to the separate latent dynamics (Model 4). In contrast, the combination of meta-learning and separate latent dynamics (Model 5) achieved significant reduction of RMSE, and the addition of progressive training strategy (Model 6) further improved this prediction, highlighting the most important innovation within ASIDE: the use of meta-learning from history to separately adapt the intrinsic and interventional dynamics, with additional benefits from progressive learning.

**Computational cost:** Table 15 in Appendix D compares the parameter count and computation time of ASIDE *vs*. representative baselines. The larger parameter count of ASIDE is due to its nature of 1) having separate models for intrinsic and each interventional dynamics, and 2) having separate meta-encoder for extracting dynamics-specific embeddings. As a result, it incurred the most significant training cost among baselines; when further performance improvement is desired, the use of progressive learning will incur another $\times 3$ computation. However, test-time efficiency of ASIDE is in between RNN/CDE and transformer based baselines. In other words, compared to baselines, ASIDE uses additional investment in training computation, but gaining significantly improved forecasting performance at a comparable inference efficieny.

## 5 CONCLUSIONS & DISCUSSION

We present ASIDE, a novel framework for learning adaptive and separable interventional dynamics with progressive meta-learning. In contrast to black-box based approaches for modelling, this approach leverages inductive biases and meta-learning based approaches to learn latent dynamics and adapt it to pass history. We validate our models on simulation data along with a real world dataset. The major contributions were also validated using an ablation study.

This model does have some limitations. The model was not designed with a specific component to tackle treatment selection biases. Although its performance remained strong at the presence of time-varyig confoudning in the presented experiments, incorporating challenges brought forth by such biases is a future work necessary. The identifiability results of meta-learning methodology in probabilistic generative models lends strong theoretical underpinning for the performance gain demonstrated by ASIDE. Although conceptually reasoned and empiricially examined in this paper, a rigorous theoretical deep dive into this direction is an exciting future avenue. Finally, while ASIDE is good at handling heterogeneity across samples, heterogeneity might also be caused by parameters of the dynamics changing with time. Handling such heterogeneity is another important next step.

## REPRODUCABILITY STATEMENT

We present the design of network architectures and training details used in our proposed method in Appendix B.2. Moreover, we provide reference to the official code repositories employed for experimentation with our baselines in Apendix G. The data simulation environment are discussed in Appendix B.1 and code is available along with baselines in G.

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

## A   SUMMARY OF MATHEMATICAL NOTATIONS

| Symbol | Definition |
|---|---|
| $\mathbf{x_t}$ | Observed covariates at time $t$ |
| $\mathbf{z_t}$ | Latent state at time $t$ |
| $\mathbf{a_t}$ | External intervention(s) applied at time $t$ |
| $\mathbf{y_t}$ | Observed outcome at time $t$ |
| $f_\theta(\mathbf{z_t}, \mathbf{a_t})$ | Composite latent dynamics function |
| $\hat{\mathbf{z}}_t$ | Estimated latent state from encoder given short past history |
| $\hat{\mathbf{y}}_t$ | Predicted outcome at time $t$ |
| $f_{\theta_{\text{int}}}(\mathbf{z}_t; \mathbf{c}_{\text{int},t})$ | Intrinsic dynamics function |
| $f_{\theta_{\text{ext}}^k}(\mathbf{z}_t, a_t^k; \mathbf{c}_{\text{ext},t}^k)$ | Effect of intervention $k$ on dynamics |
| $\mathbf{c}_{\text{int},t}$ | Context embedding for intrinsic dynamics |
| $\mathbf{c}_{\text{ext},t}^k$ | Context embedding for intervention $k$ |
| $\mathcal{I}^k(a_t^k)$ | Indicator function for intervention $k$ at time $t$ |
| $g_\eta(\mathbf{z}_t, \mathbf{a}_t)$ | Emission function to observable outcome |
| $\tau$ | Forecasting horizon length |
| $\mathcal{H}_{1:t}$ | History of covariates, interventions, and outcomes up to time $t$ |
| $\mathbf{s}_{\text{int}}$ | History segment without interventions |
| $\mathbf{s}_{\text{ext}}^k$ | History segment with intervention $k$ |
| $\mathbf{s}_{\text{ext,CF}}^k$ | Counterfactual segment without intervention $k$ |
| $\Psi_{\phi_z}$ | Neural encoder function for initial latent states |
| $\Psi_{\phi_{\text{int}}}$ | Neural encoder function for intrinsic dynamics |
| $\Psi_{\phi_{\text{ext}}^k}$ | Neural encoder function for external intervention |
| $\mathcal{M}_{\text{int}}$ | Meta-encoder for intrinsic dynamics |
| $\mathcal{M}_{\text{int}}^k$ | Meta-encoder for intervention type $k$ |
| $V_t$ | Tumor volume at time $t$ |
| $\rho$ | Intrinsic tumor growth rate |
| $K$ | Carrying capacity of the tumor growth model |
| $\beta_c$ | Effect of chemotherapy |
| $C_t$ | Chemotherapy dose at time $t$ |
| $\alpha$ | Linear effect of radiotherapy |
| $\beta$ | Quadratic effect of radiotherapy |
| $r_t$ | Radiotherapy dose at time $t$ |
| $\epsilon$ | Random noise term in tumor dynamics |

Table 5: Summary of Mathematical Notations

# B    DETAIL OF EXPERIMENTS AND ADDITIONAL RESULTS

## B.1    DATASET DETAILS

Geng et al. (2017) provide the dynamic model for the growth of non-small cell lung cancer under radiotherapy and chemotherapy interventions. This model is based on assuming that the intrinsic growth follows a Gompertz growth model. Similarly, effect of radiothrapy is taken to be linear quadratic (LQ) model and for chemotherapy is a log-cell kill model. This results in the trajectory of the tumor volume $V$ to be defined by a differential equation defined as:

$$\frac{dV}{dt} = V * (\rho \log(\frac{K}{V}) - \beta_c C(t) - (\alpha R(t) + \beta R(t)^2)) \tag{7}$$

For the growth model, Gompertz is a common way to model a more general form of logistic growth. $\rho$ represents the cell specific growth rate and $K$ is the carrying capacity. The value of $K$ is kept constant at $K = 14137.167$ across the simulation whereas $\rho$ is varied based on distribution suggested in (Geng et al., 2017) with changes for different heterogeneity setting as mentioned in 4.1.

For the chemotherapy part of the model, the value of $C(t)$ represents the drug concentration. Drug once administered is assumed to decay with an half life of $1$ timestep. based on the clinical practice of administering vinblastine at $5\ mg/m^3$ per week, the value of $C(t) = 5$ is introduced at anytime the treatment assignment dictates chemotherapy administration. The drug constantly decays with half life of $1$ timestep.

For radiotherapy, the parameters $\alpha, \beta : \alpha/\beta = 10$ are made to effect the cell volume at only the time of administration, and effects disappear immediately disappears. $R(t) = 2\ Gy$ at time of administration is kept constant to simulate practice.

### B.1.1    DISTRIBUTION OF VOLUME ACROSS HETEROGENEITY AND PROJECTION HORIZON

The distribution of volume across different stages and different projection horizon vary greatly, as shown in 4. Since the volume of ground truth are different, the RMSE calculated for these varying setting will also be different and not easily comparable. As clearly seen, volume distribution is higher for $heterogenity = 1$, thus RMSE are expected to be higher for this setting across all $\tau$.

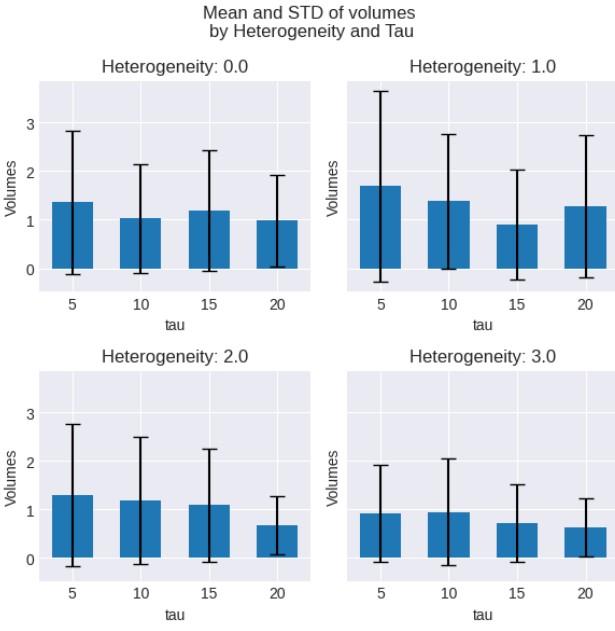

Figure 4: Volume distribution across $\tau$ for different heterogeneity

## B.2 IMPLEMENTATION DETAILS

| Component | Input | Model | Output |
|---|---|---|---|
| Latent Encoder $\Psi_{\phi_z}(\mathbf{x}_{t-l-1:t-1})$ | $x_{t-l:t-1}, a_{t-l:t-1}, l = 3$ | MLP, 3 layers | $z_t$ |
| Metamodel (intrinsic) equation 2 | $\mathbf{s}_{int}$ | MLP, 8 layers | $\mathbf{c}_{int,t}$ |
| Metamodel (intervention) equation 4 | $\mathcal{H}_{1:t}, \mathcal{I}_{1:t}^k$ | CNN, 8 layers | $\mathbf{c}_{ext,t}^{\mathbf{k}}$ |
| Dynamics (intrinsic) | $z_t, \mathbf{c}_{int,t}$ | MLP, 9 layers, 16 units | $f_{int}$ |
| Dynamics (intervention) | $z_t, \mathbf{c}_{ext,t}^{\mathbf{k}}$ | MLP, 9 layers, 16 units | $f_{ext}^k$ |
| Emission decoder | $z_t$ | MLP, 2 layers, 16 units | $y_{t+1:t+\tau}$ |

Table 6: Model components and specifications

| Encoder | | |
|---|---|---|
| **Layer** | **Output Dim** | **Details** |
| Linear + ELU | 8 | in_features = $3 \cdot F$, out_features = 8 |
| Linear + ELU | 8 | in_features = 8, out_features = 8 |
| Linear | 8 | in_features = 8, out_features = $z_{init}$ |

| Meta-encoder (intrinsic) | | |
|---|---|---|
| **Layer** | **Output Dim** | **Details** |
| Linear + ELU | $H$ | in_features = $d_{in}$, out_features = $H$ |
| (Linear + ELU)*N | $H$ | in_features = $H$, out_features = $H$ |
| Linear | $c_{int}$ | in_features = $H$, out_features = $c_{int}$ |

| Meta-encoder (intervene) | | |
|---|---|---|
| **Layer** | **Output Dim** | **Details** |
| Conv1D + ELU | $H$ | in_features = $d_{in} + d_{mask}$, out_features = $H$ |
| (Conv1D + ELU)*N | $H$ | in_features = $H$, out_features = $H$ |
| Linear | $c_{int}$ | in_features = $H$, out_features = $c_{ext}$ |

| Intrinsic Dynamics | | |
|---|---|---|
| **Layer** | **Output Dim** | **Details** |
| Linear + ELU | $H$ | in_features = $c_{int} + z_{init}$, out_features = $H$ |
| (Linear + ELU)*N | $H$ | in_features = $H$, out_features = $H$ |
| Linear | $z_t$ | in_features = $H$, out_features = $z_t$ |

| Extrinsic Dynamics | | |
|---|---|---|
| **Layer** | **Output Dim** | **Details** |
| Linear + ELU | $H$ | in_features = $c_{ext} + z_{init} + d_{int}$, out_features = $H$ |
| (Linear + ELU)*N | $H$ | in_features = $H$, out_features = $H$ |
| Linear | $z_t$ | in_features = $H$, out_features = $z_t$ |

Table 7: Architecture of the encoder, meta-encoders and dynamics networks.

## B.3 DESCRIPTION OF BASELINES

**Marginal Structural Model (MSM)** MSN is a classic model based on (Hernán et al., 2000) and extended further on (Hernan & Robins, 2020). The idea here is to learn a structured model (here linear) for first forecasting treatment from covariates: $f_{p_t}(\mathbf{a_t}|\mathbf{a_{1:t-1}})$ which is used to calculate Inverse propensity treatment weights (IPTW). These IPTW weights are used to rescale the inputs to learn another structured (again linear) model: $y_t = f_o(\mathbf{x_{t:t-1}}, \mathbf{a_{t:t-1}})$

**Recurrent Marginal Structural Network (RMSN):** Based on work by Lim (2018), RMSN is built completely on RNNs. It is a simple extension to linear marginal Structural models described in Hernan & Robins (2020) using RNN. First, the Inverse Probability Weights are learned. One RNN models: $f_{p_t}(\mathbf{a_t}|\mathbf{a_{1:t-1}})$ and another RNN models $f_{p_h}(\mathbf{a_t}|\mathbf{a_{1:t-1}}, \mathcal{H}_{1:t-1})$. Then, the weight is calculated to be: $w_i = \frac{f_{p_t} x_i}{f_{p_h} x_i}$. After learning the weights, an encoder is trained with the task of learning representations from data as: $e_{i,t} = f_e(w_i, \mathcal{H}_{1:t-1})$. The representation $e_i$ learnt above is then used to make autoregressive predictions for $y_{t+1:\tau}$ and is modeled by another RNN: $y_{i,t+1:\tau} = f_d(e_{i,t}, w_i, y_{i,t})$

**Counterfactual Recurrent Network (CRN):** CRN by Bica et al. (2020) is also an encoder-decoder architecture. However, unlike RMSN, in CRN, the bias handling is done by an adversarial loss component instead of learned weights, which reduces the need for separately modelling the weights by RNNs. The idea behind CRN, is to learn balanced representations $r_i(\mathcal{H}^i_{1:t-1})$ which is invariant to treatment assignment. This is done by building to separate heads to the RNN models, one to predict the treatment $\mathbf{a_{t+1}} = G_A(r_i(\mathcal{H}^i_{1:t-1}))$ and another to predict outcome $Y_{t+1} = G_Y(r_i(\mathcal{H}^i_{1:t-1}), \mathbf{a_t})$ at each time point. An adversarial loss is used to learn this model.

In the open source implementation provided by the Bica et al. (2020), this loss is implemented using *Gradient reversal* layer.

**Causal Transformer (CT):** CT (Melnychuk et al., 2022) extends the idea of CRN using transformer to make the pipeline end-to-end and replace the two-stage learning process in previous models. They use multi-headed self and cross-attention mechanisms common in transformer based approaches to model the sequences, which then learns to attend to different part of history to make future predictions. As in CRN, they learn the transformer by an adversarial component on top of transformer. The transformer is tasked to learn a balanced represenattion $r_i(\mathcal{H}^i_{1:t-1})$ such that it is invariant of treatment predictions. Two heads are added on top of transformer: one to predict the treatment $\mathbf{a_{t+1}} = G_A(r_i(\mathcal{H}^i_{1:t-1}))$ and another to predict outcome $Y_{t+1} = G_Y(r_i(\mathcal{H}^i_{1:t-1}), \mathbf{a_t})$ at each time point. An adversarial loss known as Counterfactual Domain Confusion (CDC) loss is used to learn the balanced representation.

**TE-CDE:** (Seedat et al., 2022) is similar in concept to CRN and CT for the most part, but for the base architecture, they differ by using a Controlled Differential Equation (CDE) following (Kidger et al., 2020). The major difference between our work and use of controlled differential equation is that CDEs require to be "controlled" by a observation pathway which in turn makes the model inherently good for reconstruction tasks but not so much for forecasting tasks. This makes the CDE models equivalent to continous latent space extensions to the Recursive family networks such as RNN, LSTMs. This continous nature helps in solving the problem of informative treatment sampling presented in Seedat et al. (2022), but becomes much less relevant when data is regularly sampled. Also, the inherent need to have a control pathway, make these models not very reliable for forecasting tasks. Like CRN, CT, these models also work on a two-stage encoder-decoder acrchitecture.

**SCIP:** Hess & Feuerriegel (2025) takes inspirarion from TE-CDE for using CDEs, but unlike the two stage architecture and domain-invariance for bias elimination, they do a single stage end-to-end training of the encoder-decoder. For bias elimination, they have a second CDE which is trained to predict treatment assignment (similar to RMSN and MSM) and use it to calculate IPTW weights for weighting the dataset.

## C  ADDITIONAL RESULTS

### C.1  FULL RESULTS:

| Heterogenity | Model | $\tau = 5$ | $\tau = 10$ | $\tau = 15$ | $\tau = 20$ |
|---|---|---|---|---|---|
| 0 | ASIDE | 0.32 (0.05) | 0.23 (0.01) | 0.31 (0.02) | 0.25 (0.01) |
| | RMSN | 0.76 (0.26) | 0.91 (0.61) | 1.74 (0.48) | 1.61 (0.92) |
| | CRN | 0.62 (0.09) | 0.69 (0.10) | 0.93 (0.25) | 1.11 (0.05) |
| | CT | 0.87 (0.21) | 0.87 (0.15) | 1.20 (0.24) | 1.02 (0.15) |
| | MSM | 0.95 | 1.25 | 2.10 | 2.10 |
| | ACTIN | 0.66 (0.35) | 0.97 (0.63) | 1.21 (0.93) | 1.27 (1.30) |
| | SCIP | 3.81 (0.73) | 5.72 (1.92) | 8.44 (1.92) | 17.83 (13.53) |
| | TE-CDE | 2.82 (0.31) | 3.41 (0.25) | 4.32 (0.62) | 3.96 (1.10) |
| | TESAR-CDE | 2.25 (0.03) | 1.99 (0.05) | 2.33 (0.11) | 1.95 (0.10) |
| 1 | ASIDE | 0.80 (0.16) | 0.59 (0.04) | 0.39 (0.05) | 0.52 (0.01) |
| | RMSN | 1.80 (0.53) | 1.85 (0.37) | 1.38 (0.65) | 1.89 (1.02) |
| | CRN | 1.35 (0.06) | 1.19 (0.02) | 0.98 (0.29) | 1.21 (0.13) |
| | CT | 1.57 (0.12) | 1.69 (0.16) | 1.11 (0.12) | 1.66 (0.20) |
| | MSM | 1.48 | 2.15 | 2.05 | 3.20 |
| | ACTIN | 1.05 (0.13) | 1.20 (0.23) | 1.42 (0.33) | 1.74 (0.37) |
| | SCIP | 3.41 (0.67) | 5.66 (2.88) | 4.54 (1.53) | 4.27 (0.87) |
| | TE-CDE | 3.42 (0.50) | 9.21 (6.02) | 8.00 (2.72) | 7.71 (1.84) |
| | TESAR-CDE | 3.18 (0.04) | 2.76 (0.05) | 2.28 (0.07) | 3.15 (0.15) |
| 2 | ASIDE | 0.66 (0.12) | 0.60 (0.03) | 0.58 (0.07) | 0.39 (0.02) |
| | RMSN | 1.19 (0.06) | 1.40 (0.33) | 1.29 (0.26) | 1.38 (0.34) |
| | CRN | 1.09 (0.04) | 1.20 (0.14) | 1.15 (0.22) | 1.20 (0.33) |
| | CT | 1.31 (0.29) | 1.37 (0.19) | 1.34 (0.17) | 1.29 (0.18) |
| | MSM | 1.14 | 1.59 | 2.04 | 1.82 |
| | ACTIN | 1.16 (0.09) | 1.28 (0.25) | 1.59 (0.47) | 2.00 (0.67) |
| | SCIP | 3.98 (0.86) | 4.37 (0.74) | 5.89 (1.73) | 7.50 (6.75) |
| | TE-CDE | 3.46 (0.33) | 9.89 (5.87) | 51.29 (46.55) | 15.42 (20.80) |
| | TESAR-CDE | 2.31 (0.01) | 2.55 (0.02) | 2.52 (0.10) | 1.89 (0.06) |
| 3 | ASIDE | 0.68 (0.02) | 0.57 (0.05) | 0.35 (0.01) | 0.37 (0.02) |
| | RMSN | 1.14 (0.03) | 1.32 (0.22) | 1.00 (0.11) | 1.23 (0.18) |
| | CRN | 1.16 (0.02) | 1.26 (0.11) | 1.02 (0.07) | 1.06 (0.13) |
| | CT | 1.29 (0.06) | 1.44 (0.11) | 1.35 (0.11) | 1.45 (0.13) |
| | MSM | 1.22 | 1.77 | 2.31 | 3.38 |
| | ACTIN | 1.31 (0.22) | 1.56 (0.32) | 1.84 (0.41) | 2.39 (0.51) |
| | SCIP | 3.51 (0.49) | 6.62 (2.14) | 5.60 (2.08) | 16.85 (10.11) |
| | TE-CDE | 3.19 (0.79) | 5.96 (2.50) | 5.15 (2.23) | 7.04 (3.70) |
| | TESAR-CDE | 2.63 (0.03) | 2.65 (0.04) | 1.62 (0.03) | 1.71 (0.03) |

Table 8: RMSE for different heterogeneity levels and projection horizon in tumor dataset.

A notable observation in Table 8 was that for most baselines, their forecasting accuracy actually improved as $\tau$ or the heterogeneity level increased (from level 1 to level 2-3). This was potentially because a higher heterogeneity level in treatment effect parameters or time-window $\tau$ have resulted in a dataset with smaller tumor volumes (and hence small RMSE) (see Fig. 4 in Appendix B.1.1).

## C.2 SOME SAMPLE VISUALS

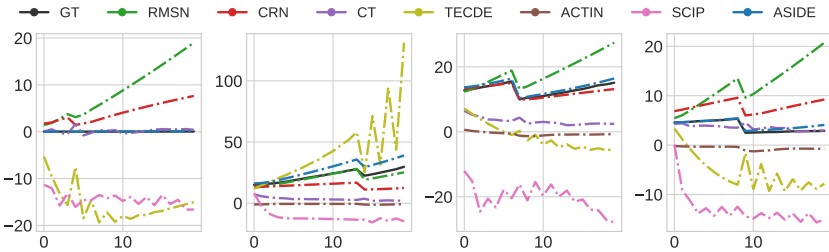

Figure 5: Some visual examples of forecast data for the different models

## C.3 SEPARATE RMSE

A summary comparison of separate RMSE across intrinsic dynamics and extrinsic dynamics for settings with heterogenity level 3 and projection horizon 20 is presented in 9. A more detailed analysis across settings can be found in table 10 for RMSN (Lim, 2018), table 11 for CRN (Bica et al., 2020), table 12 for CT (Melnychuk et al., 2022), and table 13 for ASIDE

| RMSE on | RMSN | CRN | CT | ASIDE |
|---|---|---|---|---|
| intrinsic growth | 1.30 (0.18) | 1.15 (0.13) | 1.14 (0.17) | 0.42 (0.02) |
| chemo steps only | 1.21 (0.18) | 1.04 (0.10) | 1.55 (0.12) | 0.44 (0.03) |
| radio steps only | 1.50 (0.25) | 1.38 (0.19) | 1.65 (0.22) | 0.44 (0.02) |

Table 9: Separate RMSEs for different dynamics for all models on datasets with heterogeneity level 3 and a $\tau$ of 20

| RMSE Type | Heterogeneity | 5 | 10 | 15 | 20 |
|---|---|---|---|---|---|
| Growth | 0 | 0.86 (0.30) | 1.01 (0.70) | 1.96 (0.54) | 1.78 (1.03) |
| | 1 | 2.03 (0.64) | 2.11 (0.44) | 1.52 (0.74) | 2.16 (1.18) |
| | 2 | 1.30 (0.08) | 1.58 (0.38) | 1.42 (0.30) | 1.55 (0.39) |
| | 3 | 1.20 (0.03) | 1.47 (0.27) | 1.07 (0.12) | 1.30 (0.18) |
| Chemo | 0 | 0.84 (0.30) | 0.95 (0.67) | 1.86 (0.52) | 1.72 (1.05) |
| | 1 | 1.58 (0.71) | 1.85 (0.45) | 1.39 (0.74) | 1.97 (1.22) |
| | 2 | 0.87 (0.12) | 1.34 (0.52) | 1.36 (0.36) | 1.46 (0.41) |
| | 3 | 0.73 (0.07) | 1.20 (0.38) | 1.02 (0.15) | 1.21 (0.18) |
| Radio | 0 | 0.86 (0.30) | 0.97 (0.67) | 1.96 (0.54) | 1.80 (1.11) |
| | 1 | 3.02 (0.54) | 2.85 (0.45) | 1.87 (0.77) | 2.59 (1.10) |
| | 2 | 1.98 (0.09) | 2.11 (0.24) | 1.70 (0.28) | 1.85 (0.34) |
| | 3 | 2.04 (0.05) | 2.09 (0.11) | 1.23 (0.13) | 1.50 (0.25) |

Table 10: Separate RMSE for the different segments for RMSN

| RMSE Type | Heterogeneity | 5 | 10 | 15 | 20 |
|---|---|---|---|---|---|
| Growth | 0 | 0.68 (0.10) | 0.73 (0.12) | 1.04 (0.28) | 1.24 (0.05) |
| | 1 | 1.49 (0.06) | 1.36 (0.04) | 1.10 (0.34) | 1.39 (0.17) |
| | 2 | 1.18 (0.04) | 1.34 (0.14) | 1.29 (0.26) | 1.36 (0.38) |
| | 3 | 1.22 (0.03) | 1.40 (0.12) | 1.10 (0.08) | 1.15 (0.13) |
| Chemo | 0 | 0.65 (0.12) | 0.72 (0.14) | 0.91 (0.28) | 1.18 (0.15) |
| | 1 | 0.99 (0.09) | 0.98 (0.04) | 0.94 (0.38) | 1.16 (0.18) |
| | 2 | 0.77 (0.07) | 1.00 (0.23) | 1.19 (0.35) | 1.16 (0.39) |
| | 3 | 0.77 (0.06) | 1.08 (0.15) | 1.06 (0.11) | 1.04 (0.10) |
| Radio | 0 | 0.64 (0.11) | 0.72 (0.14) | 0.99 (0.30) | 1.23 (0.11) |
| | 1 | 2.53 (0.15) | 2.19 (0.07) | 1.38 (0.14) | 1.96 (0.22) |
| | 2 | 1.78 (0.06) | 1.84 (0.10) | 1.54 (0.20) | 1.67 (0.34) |
| | 3 | 2.06 (0.03) | 2.04 (0.04) | 1.24 (0.10) | 1.38 (0.19) |

Table 11: Separate RMSE for the different segments for CRN

| RMSE Type | Heterogeneity | 5 | 10 | 15 | 20 |
|---|---|---|---|---|---|
| Growth | 0 | 1.00 (0.24) | 0.98 (0.17) | 1.36 (0.27) | 1.14 (0.17) |
| | 1 | 1.80 (0.14) | 1.96 (0.18) | 1.27 (0.14) | 1.95 (0.24) |
| | 2 | 1.43 (0.34) | 1.56 (0.23) | 1.53 (0.19) | 1.49 (0.21) |
| | 3 | 1.38 (0.07) | 1.63 (0.13) | 1.49 (0.12) | 1.63 (0.15) |
| Chemo | 0 | 0.95 (0.26) | 0.97 (0.18) | 1.34 (0.27) | 1.13 (0.15) |
| | 1 | 1.32 (0.15) | 1.67 (0.18) | 1.11 (0.13) | 1.79 (0.22) |
| | 2 | 1.10 (0.27) | 1.29 (0.20) | 1.28 (0.21) | 1.37 (0.18) |
| | 3 | 0.92 (0.07) | 1.34 (0.13) | 1.46 (0.10) | 1.55 (0.12) |
| Radio | 0 | 0.98 (0.22) | 1.00 (0.18) | 1.39 (0.29) | 1.16 (0.16) |
| | 1 | 2.54 (0.18) | 2.47 (0.40) | 1.37 (0.20) | 2.23 (0.44) |
| | 2 | 2.11 (0.38) | 2.02 (0.20) | 1.67 (0.19) | 1.73 (0.16) |
| | 3 | 2.19 (0.08) | 2.18 (0.10) | 1.50 (0.18) | 1.65 (0.22) |

Table 12: Separate RMSE for the different segments for CT

| RMSE Type | Heterogeneity | 5 | 10 | 15 | 20 |
|---|---|---|---|---|---|
| Growth | 0 | 0.36 (0.06) | 0.26 (0.01) | 0.35 (0.03) | 0.28 (0.00) |
| | 1 | 0.92 (0.22) | 0.68 (0.05) | 0.44 (0.06) | 0.60 (0.02) |
| | 2 | 0.73 (0.15) | 0.68 (0.04) | 0.65 (0.08) | 0.45 (0.02) |
| | 3 | 0.70 (0.03) | 0.64 (0.06) | 0.39 (0.01) | 0.42 (0.02) |
| Chemo | 0 | 0.37 (0.07) | 0.26 (0.01) | 0.32 (0.03) | 0.29 (0.02) |
| | 1 | 0.61 (0.16) | 0.57 (0.12) | 0.39 (0.03) | 0.53 (0.03) |
| | 2 | 0.49 (0.04) | 0.42 (0.04) | 0.50 (0.06) | 0.39 (0.02) |
| | 3 | 0.24 (0.02) | 0.47 (0.09) | 0.40 (0.02) | 0.44 (0.03) |
| Radio | 0 | 0.33 (0.03) | 0.25 (0.01) | 0.32 (0.02) | 0.29 (0.01) |
| | 1 | 1.30 (0.02) | 0.93 (0.06) | 0.55 (0.03) | 0.79 (0.04) |
| | 2 | 1.00 (0.26) | 0.99 (0.07) | 0.87 (0.17) | 0.56 (0.03) |
| | 3 | 1.38 (0.05) | 1.02 (0.13) | 0.38 (0.04) | 0.44 (0.02) |

Table 13: Separate RMSE for the different segments for ASIDE

## C.4 IDENTIFIABILITY RESULTS

| Heterogeneity | $\tau$ | $MCC(\rho)$ mean | std | $MCC(\alpha)$ mean | std | $MCC(\beta_c)$ mean | std |
|---|---|---|---|---|---|---|---|
| 1 | 5 | 0.64 | 0.03 | 0.82 | 0.02 | 0.11 | 0.02 |
| | 10 | 0.65 | 0.02 | 0.77 | 0.05 | 0.19 | 0.02 |
| | 15 | 0.65 | 0.02 | 0.80 | 0.04 | 0.15 | 0.01 |
| | 20 | 0.68 | 0.02 | 0.78 | 0.05 | 0.14 | 0.02 |
| 2 | 5 | 0.60 | 0.07 | 0.77 | 0.05 | 0.12 | 0.02 |
| | 10 | 0.69 | 0.02 | 0.81 | 0.02 | 0.15 | 0.01 |
| | 15 | 0.74 | 0.02 | 0.83 | 0.06 | 0.16 | 0.02 |
| | 20 | 0.76 | 0.02 | 0.83 | 0.03 | 0.17 | 0.03 |
| 3 | 5 | 0.82 | 0.02 | 0.87 | 0.02 | 0.14 | 0.02 |
| | 10 | 0.82 | 0.01 | 0.90 | 0.02 | 0.13 | 0.04 |
| | 15 | 0.88 | 0.01 | 0.88 | 0.02 | 0.12 | 0.01 |
| | 20 | 0.90 | 0.01 | 0.93 | 0.02 | 0.11 | 0.03 |

Table 14: MCC value for identifiability of individual parameters

## C.5 PER STEP RMSE FOR MIMIC-III

Fig 6 shows per-step RMSE for MIMIC-III

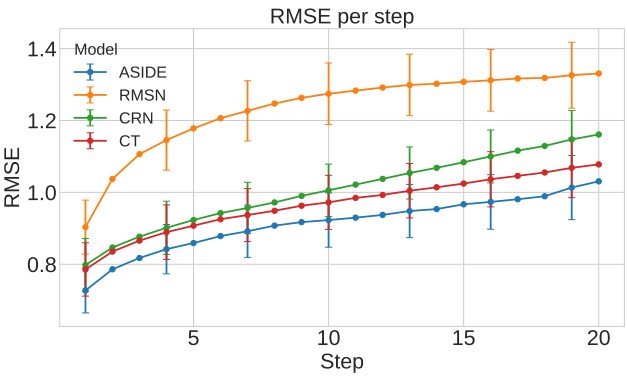

Figure 6: RMSE per step for MIMIC-III for different models

## D COMPLEXITY OF MODELS

| | CRN | CT | TESARCDE | ACTIN | ASIDE |
|---|---|---|---|---|---|
| Parameters | 7.6k | 10.1k | 5.2k | 6.1k | 32.5k |
| Train time (per epoch) | 1.62 (.84) | 17.00 (.29) | 0.54 (0.02) | 1.18 (0.03) | 55.21 (8.58) |
| Test time (per 50K seq) | 32.66 (1.03) | 114.83 (3.18) | 76.4 (1.35) | 338.2 (13) | 99.44 (7.45) |

Table 15: Comparison of parameter counts as well as training and test computation time.

## E    SOCIETAL IMPACT

This paper presents work whose goal is to advance the field of Machine Learning. There are many potential societal consequences of our work, none of which we feel must be specifically highlighted here.

## F    LLM USAGE

LLM tools such as ChatGPT, DeepSeek were used in limited capacity for refining this paper. Tasks for which LLMs were used are mostly spellchecks and grammar checks. Apart from that, LLMs were also used to generate table templates and filler codes for some plots in this paper.

## G    SOURCE CODE

Source code can be found here: `https://anonymous.4open.science/r/ASIDE/`

