# OpenReview forum: "ASIDE: Adaptive and Separable Interventional Dynamics  via Progressive Meta-Learning"
_ICLR.cc/2026/Conference — Submitted to ICLR 2026_

### Official Review · Reviewer_V4z1 · 2025-10-28

**Soundness:** 2
**Presentation:** 2
**Contribution:** 1
**Rating:** 2
**Confidence:** 4

**Summary:**

The paper addresses time‑series forecasting under external interventions. It proposes ASIDE, which decomposes predictions into baseline (intrinsic) and response (intervention‑driven) components implemented by separate networks whose outputs are combined additively. Each network takes a context embedding of the history as input, and the model is optimized for future predictive accuracy. Training proceeds in three stages: (i) intrinsic dynamics only, (ii) intervention dynamics only, and (iii) joint training. Evaluation is conducted on a synthetic dataset aligned with the method’s assumptions and on the MIMIC dataset.

**Strengths:**

* The paper targets an important problem in the healthcare domain: estimating treatment effects in medical time-series.
* On the synthetic experiment, the method substantially outperforms the baselines, however, the scenario is not realistic (see Weaknesses).

**Weaknesses:**

The paper should be rejected due to (i) limited novelty, (ii) unclear conceptual differences from prior work, (iii) focus on a special‑case problem with no clear real‑world use case, and (iv) issues with the empirical evaluation.

**(i) Limited Novelty**

The method comprises: (a) an additive decomposition with separate networks for baseline and response dynamics; (b) conditioning of these networks on history via a context embedding, and (c) a three‑stage training procedure. Here, components (a) and (b) are not novel ideas, which limits the originality of the contribution.

**(ii) Conceptual differences with prior work**

The paper does not discuss prior work relevant to (c) the three‑stage adaptive training, making it impossible to assess novelty.

In addition, several claims reference prior literature without citations, e.g.:
* Ln. 97-98: All existing intervention-effect models … - no reference.
* Ln. 99-101: … typically achieved in a two-stage encoding-decoding framework… - no reference
* Ln. 184-185: While different adaptation mechanisms exist, … - no reference
* Ln. 186-187: … differ from existing works … - no reference
* Ln. 192-194: …, existing works mostly focus on … - no reference

**(iii) Special‑case problem with no clear real‑world connection**

The paper models intervention effects without addressing time‑varying confounding, thereby tackling a special case of counterfactual outcome estimation studied in prior work [Bica+20; Melnychuk+22]. In real medical time‑series, interventions are typically assigned based on patient history. Additionally, the manuscript does not connect its restricted setting to a concrete real‑world use case.

**(iv) Issues with empirical results**

Because the method only estimates treatment effects in the absence of time‑varying confounding, the synthetic experiment uses randomly assigned treatments. This setup is unrealistic and limits the significance of the observed gains. In the real‑world MIMIC experiment, the method does not show substantial improvements over the main baseline, the causal transformer (CT) [Melnychuk+22].

**Minor comments**
* Figures and captions are not yet polished and could be improved.
* Several repeatedly used phrases are unclear and should be defined at first use:
  * “one-size-fit-all” forecasting function
  * leverage inductive bias

**Questions:**

There are several points missing in the main text:

* The training prediction horizon for the proposed method and each baseline is not reported. Please clarify the horizon used for each method in each experiment.
* Parameter counts for ASIDE and for all baselines are not provided. Please report them.
* What is the training‑time overhead of the three‑stage procedure relative to single‑stage training? Is it $3 \times$?

---

> ### Author Response · Authors · 2025-11-26
> **Response to Reviewer V4z1**
>
> **Novelty**
>
> Thanks for pointing out the need for better clarity in describing the contribution of our work. This has helped to re-examine and re-shape the key contribution of our works. We refer the reviewer to our overall response point 3 for our detailed response.
>
> **Conceptual differences with prior work**
>
> We hope our response and revision related to the innovation aspect have now clarified the question about the conceptual difference with prior works. In addition, we have added citations to all the statements  that the reviewer noted.
>
> **Addition of experiments considering time-varying confounding**
>
> Thanks for this excellent suggestion, and we refer the reviewer to our overall response point 2 for our detailed response.
>
>
> **Results on MIMIC-III**
>
> The reviewer noted that ASIDE did not show substantial improvement over the main baseline, such as CT, on MIMIC-III. We would like to clarify that 1) the margins of difference among baselines, e.g., CT over CRN, versus ASIDE over CT, were at a similar level; and 2) the improvements of ASIDE over CT were statistically significant (paired-t tests) over all prediction horizons.
>
> **Training horizons**
>
> The training prediction horizons considered \tau = 5, 10, 15, 20, and were included in all results, e.g, the x-axis in Fig 2 and Fig 3, caption in Table 2, and each column in Table 3.
>
> **Parameter counts & complexity comparison**
>
> In **Section 4.3** and **Table 15** of the revised manuscript, we compared the parameter count, training-time computation, and test-time computation for ASIDE and baselines. Yes the training-time overhead for the progressive learning is about 3x: note that in our revised ablation results (Table 4) and writing, we have clarified that the key to ASIDE’s performance gain comes from the integration of meta-learning and separate dynamics modeling to truly achieve separable and adaptive dynamics (Model 5 in Table 4). The addition of progressive training (Model 6 in Table 4) brings additional gains, but not the most important – it is a decision that can be made at training time regarding whether to investment in this 3x overhead for an additional boost of performance.

---

### Official Review · Reviewer_mjx5 · 2025-10-28

**Soundness:** 3
**Presentation:** 2
**Contribution:** 2
**Rating:** 4
**Confidence:** 4

**Summary:**

This paper proposes ASIDE (Adaptive and Separable Interventional Dynamics), a framework that disentangles latent intrinsic dynamics from interventional responses in time-series data. The method leverages (1) explicit decomposition of latent ODE components into intrinsic (with own dynamic) and intervention-dependent (influenced by action) (2) a progressive meta-learning scheme that first learns intrinsic dynamics from intervention-free segments, then adapts interventional components via counterfactual generation. Context embeddings extracted from past history enable adaptation to inter- and intra-subject heterogeneity. Experiments on synthetic and real world dataset show improved forecasting accuracy over RMSN, CRN, and CT baselines.

**Strengths:**

1. The explicit separation of intrinsic and interventional dynamics is intuitive and provides interpretability benefits over existing black-box time-series models.

2. The two staged optimization that first learns intrinsic and then interventional dynamics is a neat idea that mitigates entanglement issues common in neural ODE frameworks. (although not sure why this is framed as meta- learning but okay)

3. Both synthetic and real-world data are evaluated, with ablation studies that show each component’s contribution. In general, the empirical results are fare.

**Weaknesses:**

1. The paper is dense and difficult to follow. Key intuitions behind the meta-learning and counterfactual embedding extraction could be explained more clearly, intuitively and visually. Can you try to add more explanations?

2. While the work uses causal terminology and very much close to causal literature, it does not formally define or justify causal assumptions (especially around identifiability). Can you provide some justification or insight here?

3. Maybe some baselines are missing, e.g. causal time series models, or recent counterfactual CDE or transformer-ODE baselines. Maybe there are some other baselines to compare to?

4. Not sure how much novelty this idea have, decomposable models for time series separate default condition and conditions with intervention is not new. The literature should not just limited to neural ODE?

**Questions:**

Please see my weakness section.

---

> ### Comment · Reviewer_mjx5 · 2025-11-25
> **Rebuttal missing?**
>
> Hi, just checking if the authors are planning to submit any rebuttal for this paper?

---

> > ### Author Response · Authors · 2025-11-25
> >
> > Yes! Thanks for checking -- we're working hard to incorporate the reviewers' response in our revision. We are aiming to upload our revision and responses tomorrow. Thanks for your patience!

---

> ### Author Response · Authors · 2025-11-26
> **Response to Reviewer mjx5**
>
> **Explanation about meta-learning and counterfactual embedding extraction**
>
> We apologize for the lack of clarity in the original manuscript. As summarized in our overall response point 3, we have revised our manuscript to better summarize the intuition behind our key innovations.
>
> Specifically, the meta-learning strategy considers observed samples of \tau window in the past history, and attempts to extract embedding from these multiple examples to adapt the forecasting dynamics function. In this way, different embeddings can be extracted from different history, allowing the forecasting function to change from patient to patient.
>
> The counterfactual embedding extraction is motivated to overcome the challenge that an observed trajectory of a system under intervention shows the composite effect of the intrinsic dynamics of the system along with its response to interventions. To separate the latter part out from this composite observation, we provide a counterfactual to the observed trajectory that is intervention free – this way, the model can contrast the intervention-free versus interventional trajectory, and extract an embedding that separates out the interventional effect only. This counterfactual example can either be obtained in a paired fashion by synthesizing an intervention-free trajectory using the intrinsic dynamics that is already learned, or an unpaired fashion using factual intervention-free segments from other parts of the history.
>
> **Theory and identifiability**
>
> Thanks for this insightful feedback and we refer the reviewer to our overall response 4 for details.
>
>
> **Baselines**
>
> As summarized in our overall response, we added substantial experimental results on five new baselines covering the latest publications in this domain with an open implementation. We refer the reviewer to our overall response point 1 for more details.
>
> In addition, we  would like to thank the reviewer for suggesting potential new baselines of causal time-series models. An important distinction of dynamic intervention modeling is that the model needs to take a **growing history** of data (covariates, treatment, and outcome) to predict the future trajectory. The type of time-series models that considers a fixed look-back window therefore do not apply. Similarly, causal time series models are typically focused on reconstruction (and identification of latent variables) of time-series of fixed-lengths, which make them not applicable for taking in varying lengths of history data to predict forward. The reviewer’s suggestion however did help us realize that we should incorporate an additional review of related works to incorporate both more general time-series models and causal time series models, to clarify their distinction to dynamic interventional models. This was added to the section of Related Works.
>
> **Innovation**
>
> We refer the reviewer to our overall response point 3 for details. More specifically for the point of decomposing time series into default condition and those under intervention, we would like to clarify that 1) it is relatively new in the recent ML/DL literature of intervention-effect modeling over time (as reflected in the baselines), where this mechanism of separate modeling is only considered in one prior work (IMODE), 2) the decomposed modeling alone is not the key reason for the success of ASIDE, as it does not ensure the separate model components can be separately inferred, and 3) it’s the integration of the meta-learning strategy and this model decomposition that allows separable and adaptive latent dynamics, which is the key innovation of ASIDE. Empirical evidence for these points was added to our revised Ablation results (Table 4).

---

### Official Review · Reviewer_yo1o · 2025-10-31

**Soundness:** 2
**Presentation:** 3
**Contribution:** 3
**Rating:** 6
**Confidence:** 3

**Summary:**

This paper proposes a new framework, ASIDE, for modeling time-series data under external interventions. ASIDE is designed to explicitly separate the intrinsic system dynamics and responses to external interventions at the latent space. To address inter- and intra-subject variability, ASIDE leverages a meta-learning approach, extracting context embeddings from historical data to adapt both the intrinsic and intervention dynamics. This design allows the model to handle heterogeneity across and within subjects, thereby improving forecasting accuracy, especially over longer prediction horizons and with increasing heterogeneity. The framework is evaluated on both synthetic and real-world datasets to demonstrate its effectiveness.

**Strengths:**

1. The motivations for disentangling and adapting the intrinsic and intervention-driven dynamics in time-series data are clearly described, and corresponding solutions are proposed.
2. Experiments on both synthetic and real-world datasets, including ablation studies, are conducted to demonstrate ASIDE’s advantages.
3. The paper is generally well-written and easy to follow.

**Weaknesses:**

1. The baselines are too few (only three) and outdated (the newest is from 2022). The authors should consider more advanced baselines. In addition, classical non-neural-network-based models are lacking.
2. The proposed model appears to require more computational complexity due to its more complicated structure and training process. A complexity comparison should be provided and discussed.
3. While the model is claimed to be more interpretable, the paper does not provide qualitative or quantitative analyses of interpretability benefits.

**Questions:**

Same as the Weaknesses.

---

> ### Author Response · Authors · 2025-11-26
> **Response to Reviewer yo1o**
>
> **Baselines**
>
> As summarized in our overall response, we added substantial experimental results on five new baselines covering the latest publications in this domain with an open implementation. This includes MSM to represent classical non-neural-network-based models as suggested by the reviewer. We refer the reviewer to our overall response point 1 for more details.
>
> **Complexity comparison**
>
> In **Section 4.3** and **Table 15** of the revised manuscript, we compared the parameter count, training-time computation, and test-time computation for ASIDE and baselines.
>
>
> **Interpretability**
>
> By interpretability, we were referring to the fact that the latent dynamics function in ASIDE has explicit components that describe the intrinsic dynamics of a system and separate components that describe the effect of each external intervention. This formulation is more interpretable than a single black-box model that does not have these separate components. We agree that we should avoid emphasizing this aspect of the model without further empirical, and have reduced the reference to “interpretability” throughout the revised manuscript.

---

### Official Review · Reviewer_XZKz · 2025-11-02

**Soundness:** 3
**Presentation:** 3
**Contribution:** 2
**Rating:** 6
**Confidence:** 3

**Summary:**

The authors propose a framework called Adaptive and Separable Interventional Dynamics (ASIDE) to address challenges in modeling dynamic systems and time series forecasting. Using both synthetic and real-world benchmarks, they demonstrate that ASIDE improves forecasting accuracy for both intrinsic and interventional dynamics, under settings with or without time-varying confounders.

**Strengths:**

The paper is well written, and the proposed method is conceptually clear and easy to follow.
The authors conduct experiments on both synthetic and real-world datasets, providing empirical evidence for the effectiveness of ASIDE.

**Weaknesses:**

Several concerns should be addressed to strengthen the paper:

a. Outdated baselines:
Most baseline methods used for comparison are from before 2022. More recent baselines, such as [1] and [2], should be included to better demonstrate the advantages of ASIDE.

b. Lack of theoretical justification:
The theoretical support for the method is insufficient. The authors should provide a more rigorous theoretical analysis or justification to explain why ASIDE can outperform existing methods.

c. Limited datasets:
Only two datasets are used in the experiments. Additional datasets, especially those focused on out-of-distribution (OOD) forecasting, should be included to provide a more comprehensive evaluation.

d. Minor contribution of meta-learning:
The discussion of meta-learning in the paper appears somewhat trivial and should not be emphasized as a major contribution. This component represents a natural approach for representation learning in time series forecasting rather than a novel idea.

References

[1] Liu, Haoxin, et al. "Time-series forecasting for out-of-distribution generalization using invariant learning." arXiv preprint arXiv:2406.09130 (2024).

[2] Wang, Yuxuan, et al. "Timexer: Empowering transformers for time series forecasting with exogenous variables." Advances in Neural Information Processing Systems 37 (2024): 469–498.

**Questions:**

Please refer to the weaknesses above. Addressing these issues would significantly strengthen the paper and clarify the contribution of the proposed method.

---

> ### Author Response · Authors · 2025-11-26
> **Response to Reviewer XZKz**
>
> We thank the reviewer for the support of our original manuscript, and we address the reviewer’s previous comments below.
>
> **Baselines (a) and Datasets (c)**
>
>  As summarized in our overall response, we added substantial experimental results including 1) five new baselines covering the latest publications in this domain with an open implementation, and 2) a new experimental setting considering increasing levels of time-varying confounding in the synthetic dataset. We refer the reviewer to our overall response (points 1-2) for more details.
>
> For the datasets used, we’d like to clarify that the scope of the dataset used in our revision – the synthetic tumor growth dataset and real MIMIC-III – now represents the state-of-the-art in the literature of modeling intervention effect over time (e.g., see ACTIN 2024, TESAR-CDE 2025, and all baseline works included).
>
> We also would like to thank the reviewer for suggesting the potential new baselines. An important distinction of dynamic intervention modeling is that the model needs to take a **growing history** of data (covariates, treatment, and outcome) to predict the future trajectory. The type of time-series models that considers a fixed look-back window, including the examples suggested by the reviewer, therefore do not apply. This however did help us realize that we should incorporate an additional review of related works about more general time-series models and clarify their distinction to dynamic interventional models – this was added to the section of Related Works.
>
>  **Theoretical justification (b)**
>
> Thanks for this insightful feedback and we refer the reviewer to our overall response 4 for details.
>
> **Contribution of meta-learning (d)**
>
> Thanks for this insightful feedback. With respect to meta-learning in particular, in brief: 1) the use of meta-learning to extract context from history to adapt future dynamics is new in the specific problem setting of intervention effect modeling over time, where all existing works use the history to get a latent state variable at time t as the initial latent state of the forecasting dynamics function – we provided detailed analyses in Figs 2-3 to show that this strategy was important for addressing longer prediction horizons; 2) we do acknowledge that 1) is not new in the broader domain of meta-learning, and the fundamental innovation of ASIDE comes from the integration of meta-learning to use different parts of the history data as context to separately adapt the intrinsic and interventional dynamics: i.e., using intervention-free segments of the history to adapt intrinsic dynamics, and using the contrast of intervention-free and interventional segments of the history to adapt each interventional dynamics. This design is highly novel and does not exist in prior arts to the best of our knowledge. We refer the reviewer to our overall response 3 for additional elaborations.

---

### Author Response · Authors · 2025-11-26
**Summary of Major Revision**

We thank all reviewers for their support and constructive feedback. Below we summarize the major changes included in the revised manuscript, in response to previous comments shared across reviewers. Additional responses to individual reviewers are posted in the later sections.

**Addition of baselines** (all reviewers):

   We thank the reviewers for the suggestion of additional baselines.
   - Many of the suggested works were not suitable as a baseline: to clarify, we added a section in Related Works under **General time series modeling** to better summarize the key points in which longitudinal intervention effect modeling distinguishes from the large body of literature in time-series modeling/forecasting, primarily in its need to consider external control inputs and increasing/varying-length history data, which cannot be easily accommodated by general time-series models.
   - In the meantime, we did expand our suite of baselines: 1\) we added **MSM** to represent classical non-neural-network baseline, as suggested by Reviewer yo1o; 2\) we refreshed our literature review and included more recent dynamic intervention models with open implementations, including **TE-CDE** (ICML 2022), **TESCAR-CDE** (ICML 2023), **ACTIN** (ICML 2024), and **SCIP** (ICLR 2025). The results of these **five additional baselines** were added to all experiments in the revised manuscript, including the new experimental setting considering confounding as described below.

**Addition of experiments with time-varying confounding** (Reviewer V4z1):


   - We would like to first reiterate a key motivation of ASIDE: indeed, the vast majority of existing works in dynamic intervention modeling have focused on the important challenge of treatment assignment bias in observational data. This however leaves a missed opportunity to improve the strategies in dynamic modeling and inference, a gap of research ASIDE is motivated to fill. Experimental settings without confounding allows us to isolate and demonstrate that this component is critical for improving dynamic outcome prediction.
   - In the meantime, we do agree with the reviewer that testing ASIDE in experimental settings with confounding will further demonstrate its real-world use. We therefore **performed additional experiments on the synthetic tumor datasets considering varying levels of time-varying confounding**, in which we tested ASIDE with and without being enriched with the common approach of invariant-representations to address confounding. Experiments were added to **Section 4.1.2 and Table 2** in the revised manuscript, in which we show that the performance gain of ASIDE over baselines remained in this more challenging experimental setting.

---

> ### Author Response · Authors · 2025-11-26
> **Summary of Major Revision (continued)**
>
> **Clarification of innovation and relation to prior works** (Reviewer mjx5 and V4z1):
>
>    Overall, we’d like to clarify that – while both separate dynamics and meta-learning are not new in the broader literature (but new to the literature of intervention effect modeling over time ) – it is their careful combination, the use of different parts of the history data as context to separately adapt the intrinsic and interventional dynamics, that is the fundamental innovation of ASIDE. We elaborate below, with supporting data from a **revised ablation study** added to **Section 4.3 and Table 4** of the revised manuscript. ASIDE has 3 key elements:
>    - The separation of intrinsic and interventional dynamics, to our knowledge, was only considered in one other prior work (IMODE) in the ML/DL literature of intervention effect modeling over time. We do acknowledge that, in the broader literature outside this domain, separate intrinsic and interventional dynamics is not new. However, separate models do not guarantee that they can be separately inferred. As verified in our ablation study in Table 4, this component alone did not bring significant improvements over baselines.
>    - The use of meta-learning to extract context from history data and to adapt outcome prediction function, to our knowledge, is again new in the ML/DL literature of intervention effect modeling over time. We do acknowledge that the idea of using meta-learning to adapt a network is not new in the broad literature. As shown in our ablation study in Table 4, this component alone did not bring significant improvements over baselines either.
>    - The most important innovation of ASIDE is the careful integration of meta-learning to use different parts of the history data as context to separately adapt the intrinsic and interventional dynamics: *i.e.*, using intervention-free segments of the history to adapt intrinsic dynamics, and using the contrast of intervention-free and interventional segments of the history to adapt each interventional dynamics. This design is highly novel and does not exist in prior arts to the best of our knowledge. As shown in our ablation study in Table 4, this is where ASIDE was able to bring significant improvements over baselines. Progressive learning builds naturally on the above separate adaptation strategy and was able to further improve performance (but not as critical).
>
>    We added these clarifications in the revised manuscript 1\) in the revised ablation study in **Section 4.3**, and 2\) revised summary of innovation in **Introduction section**.
>
> **Theory and relation to identifiability** (Reviewer XZKz and mjx5):
>
>    Thanks for this insightful feedback. Meta-learning was recently established as a novel solution for achieving the identifiability of latent data-generating factors by allowing the construction of a conditional generative model leveraging auxiliary information of context samples (Ye et al, NeurIPS 2024). In the context of ASIDE, it is related to the identifiability of the parameters governing the latent intrinsic and interventional dynamics, which is in turn related to the independent causal mechanisms underlying the outcome of interest. Because ASIDE is formulated as a deterministic model, we did not attempt a rigorous theoretical proof in the revised manuscript but **added its discussion in Section 5** as an exciting future avenue for ASIDE. We did add this conceptual connection to the identifiability theory in **Section 3.4** of the revised manuscript, along with **empirical identifiability metrics** (mean correlation coefficients / MCC) on  intrinsic and interventional dynamic parameters to support this theoretical plausibility (discussed in **Section 4.1.1 with results in Table 15 of Appendix C4**). To our knowledge, this is the first time the identifiability of the latent dynamics being discussed in the ML/DL literature of intervention effect modeling over time.
>
> **Computation cost and parameter count** (Reviewer yo1o and V4z1):
>
>    In **Section 4.3** and **Table 15** of the revised manuscript, we compared the parameter count, training-time computation, and test-time computation for ASIDE and baselines.

---

> ### Author Response · Authors · 2025-12-03
> **Edit Dec 3: Minor changes**
>
> A minor edit was made to the manuscript in Table 15 of Appendix: Section D. The parameter count number for the models were corrected.

---

### Meta-Review · Area_Chair_orfz · 2026-01-03

**Summary:**

The main outstanding issues concern:

The lack of identifiability analysis supporting the claimed separation of intrinsic and interventional dynamics.

The resulting ambiguity in the method’s conceptual novelty relative to prior work.

The need for clearer and more rigorous problem formulation and presentation.

**Reviewer Concerns:**

Main Concerns
1. Identifiability (Reviewer mjx5)

Reviewer mjx5 raised a fundamental concern regarding the identifiability of the proposed separation between intrinsic latent dynamics and external intervention effects, which is the central motivation and design principle of the method.

In the rebuttal, the authors mainly refer to existing identifiability results from prior work to address this concern. However, this response remains insufficient. It is unclear whether the implicit generative model underlying the proposed formulation satisfies the assumptions required by those identifiability results. In particular, the paper does not explicitly specify the generative model or the conditions under which the intrinsic and interventional components would be identifiable.

Without a dedicated identifiability analysis tailored to the proposed model, it remains unclear whether the method can in fact recover or meaningfully separate the intrinsic dynamics and intervention effects, as claimed. This raises concerns about whether the separation achieved by the method reflects true underlying mechanisms or merely represents a convenient but potentially arbitrary decomposition optimized for predictive performance.

2. Novelty (Reviewers mjx5, V4z1, and XZKz)

The concern about novelty is also closely tied to the identifiability issue above.

If the intrinsic and interventional dynamics are not identifiable and remain potentially mixed, then the proposed method is not clearly different in principle from existing approaches that model composite dynamics without explicit separation. In this case, the claimed conceptual novelty of learning separable dynamics is weakened, and it becomes unclear what fundamentally distinguishes the method from prior latent dynamics models with control inputs.

Moreover, without a principled argument that the separation corresponds to meaningful latent mechanisms, it is difficult to justify why the proposed method should outperform existing baselines beyond empirical observation alone (Reviewers mjx5 and V4z1). This concern was also raised by Reviewer XZKz regarding the lack of clear explanation for the source of performance gains.

3. Presentation and Formulation (Reviewers mjx5 and V4z1)

I agree with Reviewers mjx5 and V4z1 that the presentation requires improvement.

Beyond repeated phrasing and clarity issues noted by Reviewer V4z1, there are more fundamental problems in the problem formulation and model description. For example, in Sec. 3.1 it is unclear how the observed covariates x are generated or what their assumed relationship is to the latents z. Although the introduction contains the sentence “Consider time-series covariate data x and its corresponding latent dynamics states z,” this statement is informal and does not constitute a formal generative or probabilistic assumption.

This lack of formalization makes it difficult to interpret the model, assess its assumptions, or evaluate whether quantities such as c_{int} and c_{ext} can be meaningfully identified from the data, as claimed in line 194. As a result, key claims remain conceptually unclear.

**Reviewer Scores:**

Reviewer mjx5’s primary concern is the lack of an identifiability result. While the rebuttal attempts to clarify the motivation and points to related identifiability results in the literature, it does not directly resolve the core concern, e.g., whether the assumptions required by those results apply to the proposed model. As a result, Reviewer mjx5 is unlikely to increase their score.

Reviewer V4z1’s main concerns focus on limited novelty, unclear presentation, and issues with the empirical setup and realism. The rebuttal does not convincingly establish that the proposed decomposition goes beyond a combination of existing ideas (in particular, no identifiability analysis is provided), nor does it offer a clearer connection to real-world use cases or substantially new empirical evidence addressing the realism concerns. As a result, Reviewer V4z1 is unlikely to change their overall recommendation.

---

### Decision · Program_Chairs · 2026-01-26

Reject